# STORK: Faster Diffusion and Flow Matching Sampling by Resolving both Stiffness and Structure-Dependence

**Zheng Tan**[1]    **Weizhen Wang**[2]    **Andrea L. Bertozzi**[1]    **Ernest K. Ryu**[1]
[1]Department of Mathematics, University of California, Los Angeles
[2]Department of Computer Science, University of California, Los Angeles
`{zhengtan, bertozzi, eryu}@math.ucla.edu`
`wzwang1210@cs.ucla.edu`

## Abstract

Diffusion models (DMs) and flow-matching models have demonstrated remarkable performance in image and video generation. However, such models require a significant number of function evaluations (NFEs) during sampling, leading to costly inference. Consequently, quality-preserving fast sampling methods that require fewer NFEs have been an active area of research. However, prior training-free sampling methods fail to simultaneously address two key challenges: the stiffness of the ODE (i.e., the non-straightness of the velocity field) and dependence on the semi-linear structure of the DM ODE (which limits their direct applicability to flow-matching models). In this work, we introduce the Stabilized Taylor Orthogonal Runge–Kutta (STORK) method, addressing both design concerns. We demonstrate that STORK consistently improves the quality of diffusion and flow-matching sampling for image and video generation. Code is available at `https://github.com/ZT220501/STORK`.

## 1 Introduction

Diffusion models (DMs) (Sohl-Dickstein et al., 2015; Ho et al., 2020; Song et al., 2021) have demonstrated remarkable achievements during the past several years on various tasks, including image generation (Dhariwal & Nichol, 2021; Meng et al., 2022), text-to-image generation (Ramesh et al., 2022; Rombach et al., 2022; Gu et al., 2022; Podell et al., 2023; Mo et al., 2023; Zhang et al., 2023), video generation (Ho et al., 2022; Kong et al., 2024; Jin et al., 2025; Liu et al., 2025), and diffusion policy (Chi et al., 2024). As shown in Song et al. (2021), the sampling of DMs can be treated as solving an SDE or an equivalent ODE backward in time. However, DMs require a significant number of function evaluations (NFEs) of the learned model when performing sampling, leading to costly inference.

Consequently, quality-preserving fast sampling methods have become an active area of research, and many methods that achieve effective sampling with fewer NFEs have been proposed (Song et al., 2022; Liu et al., 2022; Zhang & Chen, 2023; Lu et al., 2022; 2025; Xie et al., 2025; Zhao et al., 2023). As noted in (Liu et al., 2024a), the "straightness" of the velocity field is a key consideration in such fast sampling methods, and this observation corresponds to the notion of *stiffness* in classical numerical analysis (Burden & Faires, 2011). Building on this connection, exponential integrator techniques for stiff ODEs (Hochbruck & Ostermann, 2010) have been successfully adapted into the DPM-Solver (Lu et al., 2022) and DEIS (Zhang & Chen, 2023) samplers. However, these approaches rely on the semi-linear structure of the ODE formulation in noise-based diffusion models, and thus cannot be directly applied to flow-matching-based models. In such cases, additional approximations, such as the data prediction step, are required (Lu et al., 2025; Xie et al., 2025).

**Contribution.** In this work, we introduce a fast training-free sampler, the ***Stabilized Taylor Orthogonal Runge–Kutta (STORK)*** method, built upon stabilized Runge–Kutta (SRK) methods (Abdulle, 2002; Meyer et al., 2014; Skaras et al., 2021). As emphasized in Figure 4, STORK is a

stiff solver that does not rely on the semi-linear structure of noise-based diffusion model's ODE formulation, making it applicable to both noise-based and flow-based models. Our experiments show that STORK consistently improves the quality of diffusion and flow-matching sampling for image and video generation.

| Flow-Euler | Flow-DPM-Solver++ | Flow-UniPC | **STORK (Ours)** |

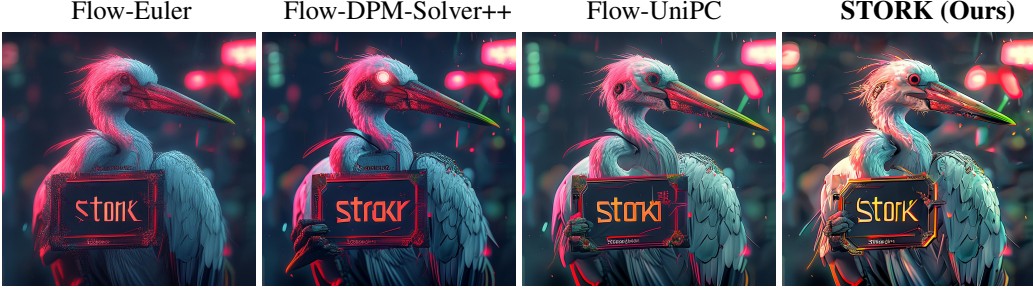

"A STORK holding a plaque with the text "STORK" on it.  Cyberpunk, 8K
resolution, highly detailed, RTX-On, super resolution"
We have the correct word with better details.

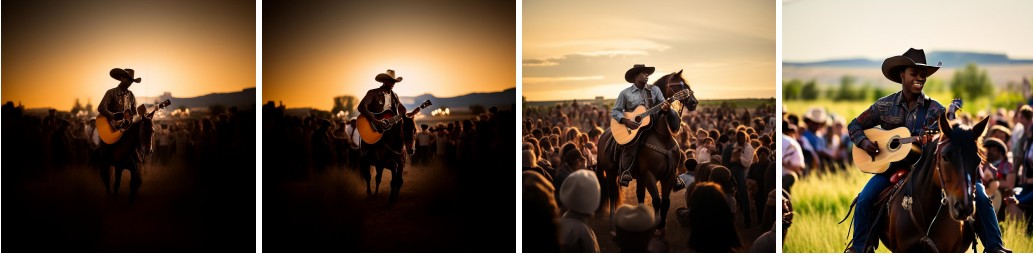

"A young black cowboy country music singer performing with a guitar on
his horse in a field outside the city to a crowd of fans, 8k, full HD"
We have a more vivid background and better realism.

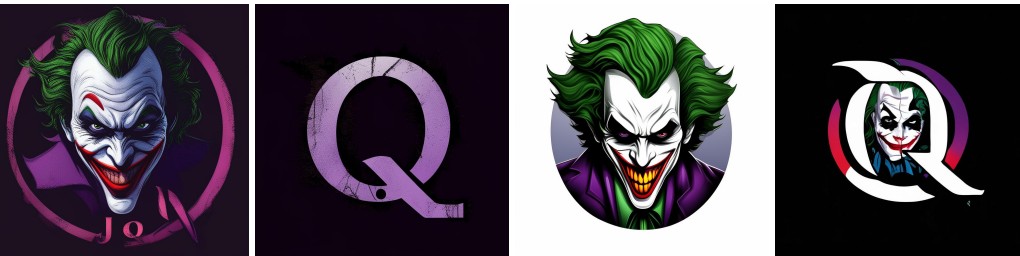

"Joker logo with the initials of the letter Q"
Only our method has both the Joker logo and the letter Q.

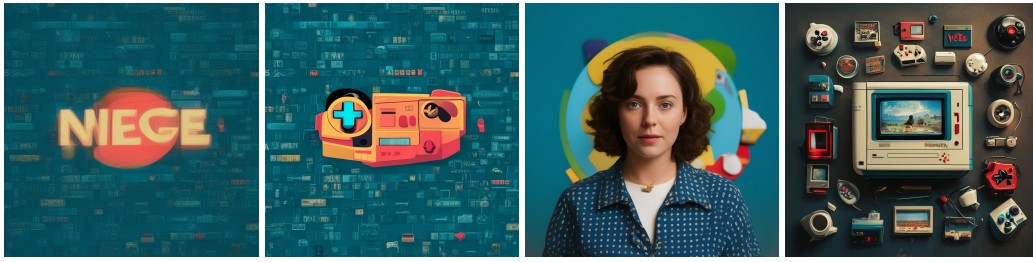

"Create an image that evokes nostalgia, using imagery of video games,
movies, and other objects that remind us of our past"
Our image is further detailed and has more diverse objects matching the prompt.

Figure 1: Comparison between the Flow-Euler, Flow-DPM-Solver++ (Lu et al., 2025; Xie et al., 2025), Flow-UniPC (Zhao et al., 2023), and STORK. All images are generated using the SANA 1.6B model (Xie et al., 2025) at $1024 \times 1024$ resolution with only 8 NFEs. Prompts are displayed beneath each image pair, accompanied by our commentary explaining why STORK's generations are superior. STORK achieves much better visual fidelity at the extremely low NFE case, showing its effectiveness as a fast sampling method. Zoom in for better visual details.

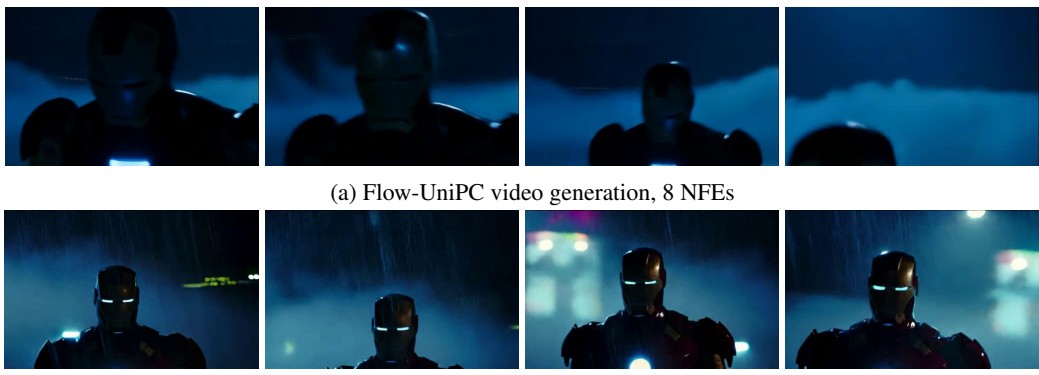

(a) Flow-UniPC video generation, 8 NFEs

(b) STORK video generation, 8 NFEs

Figure 2: Video generation on Hunyuan model (Kong et al., 2024) with prompt:
"Iron Man is walking towards the camera in the rain at night, with
a lot of fog behind him. Science fiction movie, close-up".
Our video portrays Iron Man more clearly and has rain in the background.

## 2 BACKGROUND AND RELATED WORKS

**Diffusion models.** Diffusion models (DMs) generate samples by numerically solving the reverse-time ODE

$$\frac{d\boldsymbol{x}}{dt} = f(t)\boldsymbol{x}(t) + \frac{g(t)^2}{2\sigma_t}\boldsymbol{\epsilon}_\theta(\boldsymbol{x}(t), t), \quad t \in [0, T], \tag{1}$$

where the "initial" condition $\boldsymbol{x}_T$ is sampled as a Gaussian, $f(t)$, $g(t)$, $\sigma_t$ are known functions, and $\boldsymbol{\epsilon}_\theta(\boldsymbol{x}(t), t)$ is the noise prediction model, a trained neural network. For further information on how diffusion models are trained and how they are able to generate high-quality images, we refer the readers to standard references (Sohl-Dickstein et al., 2015; Ho et al., 2020; Song et al., 2022; 2021).

Notice that equation 1 exhibits a *semi-linear* structure with a linear term $f(t)\boldsymbol{x}(t)$ and a non-linear term involving $\boldsymbol{\epsilon}_\theta$. The DPM-Solver (Lu et al., 2022) is a numerical ODE solver that exploits the semi-linear structure of equation 1, and it is one of the most widely used fast sampling methods for diffusion models.

**Flow matching objectives.** Flow matching is a class of generative models closely related to diffusion models and has recently attracted significant attention. Unlike diffusion models, which learn a noise model (score function), flow matching models directly learn a vector field $\boldsymbol{v}_\theta$ and generate samples by numerically solving the forward-time ODE

$$\frac{d\boldsymbol{x}}{dt} = \boldsymbol{v}(\boldsymbol{x}(t), t), \quad t \in [0, T], \tag{2}$$

where the initial condition $\boldsymbol{x}_0$ is sampled as a Gaussian. For further information on how flow matching models are trained and how they are able to generate high-quality images, we refer the readers to standard references (Lipman et al., 2023; Podell et al., 2023; Xie et al., 2025).

Because the flow matching ODE equation 2 lacks a semi-linear structure, the DPM-Solver (Lu et al., 2022) cannot be directly applied. Although follow-up works (Lu et al., 2025; Xie et al., 2025; Zheng et al., 2023) circumvent this limitation using a so-called data prediction step, this approach introduces errors at each step.

**Classifier-free guidance for conditional sampling.** In conditional sampling, a generative model generates samples relevant to a conditional variable $c$, and classifier-free guidance is the most widely used technique for conditional sampling with diffusion models (Dhariwal & Nichol, 2021; Ho & Salimans, 2022). Specifically, given a parametrized model $\boldsymbol{\epsilon}(\boldsymbol{x}_t, t, c)$, the conditional noise is chosen to be

$$\tilde{\boldsymbol{\epsilon}}(\boldsymbol{x}_t, t, c) = s\boldsymbol{\epsilon}(\boldsymbol{x}_t, t, c) + (1 - s)\boldsymbol{\epsilon}(\boldsymbol{x}_t, t, \emptyset),$$

where $s \geq 0$ is the classifier-free guidance-scale, $\boldsymbol{\epsilon}(\boldsymbol{x}_t, t, c)$ is the noise model conditioned on $c$, and $\boldsymbol{\epsilon}(\boldsymbol{x}_t, t, \emptyset)$ represents the unconditional noise with $\emptyset$ serving as a place holder. Then, one simply replaces the noise $\boldsymbol{\epsilon}(\boldsymbol{x}_t, t)$ in equation 1 with $\boldsymbol{\epsilon}(\boldsymbol{x}_t, t, \emptyset)$.

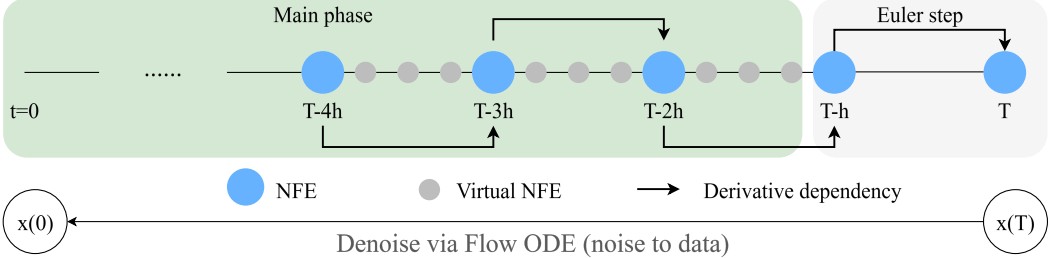

Figure 3: Illustration of NFE evaluations for STORK-4 with $s = 4$ and 1st order Taylor approximation. For presentation clarity, we use uniform timesteps with size $h$. "NFE" denotes actual NFEs, while the "virtual NFE" denotes NFEs approximated with the Taylor expansion. The arrows indicate that the previously computed velocity is used for first derivative approximations. Euler's method is used for the first step since there is no previous velocity.

For flow matching models such as Stable-Diffusion-3.5 (Esser et al., 2024), FLUX.1-dev (BlackForest, 2024), and SANA (Xie et al., 2025), a similar approach is used, replacing the velocity model $\boldsymbol{v}(\boldsymbol{x}(t), t)$ of equation 2 with a similarly defined $\tilde{\boldsymbol{v}}(\boldsymbol{x}(t), t, c)$.

**Current fast sampling methods.** Existing fast sampling methods generally fall into two categories: one needs additional training, such as knowledge distillation (Salimans & Ho, 2022; Starodubcev et al., 2025), consistency models (Song et al., 2023; Lu & Song, 2025; Chen et al., 2025), and one-step diffusion models (Liu et al., 2024a; Yin et al., 2024; Chen et al., 2025); the other is training-free, which uses various numerical solvers for Stochastic Differential Equations (SDEs) and Ordinary Differential Equations (ODEs) (Song et al., 2022; Karras et al., 2022; Jolicoeur-Martineau et al., 2021; Liu et al., 2022; Zhang & Chen, 2023; Lu et al., 2022; Zheng et al., 2023; Zhao et al., 2023; 2024). It was first shown in Song et al. (2021) that the sampling in DMs is equivalent to a backward SDE or ODE. Gotta Go Fast (Jolicoeur-Martineau et al., 2021) tried to accelerate the backward SDE sampling. EDM (Karras et al., 2022) then tried to solve the issue that the trajectory between image and noise distribution is not straight enough with better training of DMs, and used Heun's method for solving the ODE. PNDM (Liu et al., 2022) demonstrated that direct utilization of classical Runge–Kutta (RK) methods cannot result in fast sampling, therefore proposed a pseudo-numerical method on the noise manifold, using 4-step Adams-Bashforth and 4-step RK as initial steps. DEIS (Zhang & Chen, 2023) and DPM-Solver (Lu et al., 2022) were concurrent works that tried to utilize the semi-linear structure in the ODE in noise-based DMs, and used the exponential integrator (Hochbruck & Ostermann, 2010) technique to solve it. Following-up works of the DPM-Solver (Lu et al., 2025; Zheng et al., 2023; Xie et al., 2025) used data prediction to adopt the DPM-Solver into flow-based models, and introduced multi-step version of DPM-Solver. UniPC and DC-Solver (Zhao et al., 2023; 2024) proposed a predictor-corrector formulation that can be plugged into any fast sampling method in order to boost the convergence order for sampling with extremely small NFEs.

## 3 STABILIZED TAYLOR ORTHOGONAL RUNGE–KUTTA (STORK)

We now present our main method *Stabilized Taylor Orthogonal Runge–Kutta* (STORK), which is based on stabilized Runge–Kutta (SRK) methods (Verwer et al., 1990; Abdulle & Medovikov, 2001; Abdulle, 2002; Meyer et al., 2014; O'Sullivan, 2019; Skaras et al., 2021; Tan et al., 2025). SRK methods in classical numerical analysis use orthogonal polynomials such as the Chebyshev polynomial (Verwer et al., 1990; Abdulle & Medovikov, 2001; Abdulle, 2002) to mitigate the effect of stiffness (Burden & Faires, 2011) and to allow for larger timesteps. However, standard SRK methods require an excessive number of function evaluations (NFEs), so we approximate them and obtain our main methods STORK-2 and STORK-4, described fully as Algorithms 2 and 1 in Appendix E.

### 3.1 STABILIZED RUNGE–KUTTA (SRK) METHODS

Stabilized Runge–Kutta (SRK) methods is a class of novel explicit solvers in numerical analysis. To motivates the necessity of applying SRK to noise-based and flow-matching-based diffusion models, two important aspects named stiffness and structure-dependency need to be considered.

**Stiff ODEs and their solvers.** Prior works (Wizadwongsa & Suwajanakorn, 2023; Lu et al., 2022; Karras et al., 2022) have shown that the trajectory between the target distribution and the noise distribution may be "not straight enough" and that this causes challenges for fast sampling and high-quality image generation. This, in fact, corresponds to the classical notion of *stiffness* in numerical analysis (Burden & Faires, 2011). Loosely speaking, an ODE is stiff if the local change in the solution slope is excessively fast so that large timesteps produce inaccurate or even unstable solutions that blow up, causing failure of typical explicit numerical methods. One modern technique is the exponential integrator (Hochbruck & Ostermann, 2010), which is the motivation of the DPM-Solver and similar methods (Zhang & Chen, 2023; Lu et al., 2022; 2025; Xie et al., 2025). More detailed discussion of the stiffness concept can be found in Appendix B.

**Structure-dependency.** Although the methods derived from the exponential integrator technique resolves the stiffness issue, they largely depend on the semi-linear structure of the ODE of the noise-based diffusion model. More precisely, this class of methods are designed for solving stiff ODEs of the form $\frac{dx}{dt} = Lx + N(x(t))$ where $L$ is a non-zero linear operator and $N$ is the non-linear part, and therefore called *structure-dependent solvers*. Methods without requiring any special form for the ODE are, therefore, called *structure-independent solvers*. Note that the ODEs for flow-matching models do not have the semi-linear structure. Therefore, structure-dependent solvers (Lu et al., 2022; 2025; Xie et al., 2025) cannot be directly applied, and additional techniques such as data prediction need to be conducted.

**SRK methods.** We now introduce the SRK method in numerical analysis under the flow matching setting; a similar derivation holds by replacing $\boldsymbol{v}$ by $\boldsymbol{\epsilon}_\theta$ for noise-based DMs. Consider the ODE

$$\frac{d\boldsymbol{x}}{dt} = \boldsymbol{v}(\boldsymbol{x}(t), t), \quad t \in [0, T],$$

where $\boldsymbol{x} \in \mathbb{R}^d$. The SRK methods is a class of single-step, multi-stage methods that are primarily designed to allow one to numerically solve stiff ODEs with larger timesteps. In practice, the most widely used SRK methods are the second-order and fourth-order SRK methods (Abdulle & Medovikov, 2001; Abdulle, 2002; Verwer et al., 1990; Meyer et al., 2014; O'Sullivan, 2019; Skaras et al., 2021).

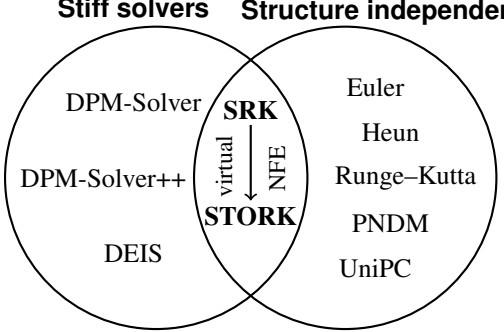

Figure 4: Derived from the SRK method, STORK is both a stiff problem solver and a structure-independent solver.

Let $h$ denote the timesteps used for solving the flow matching ODE one step from $x(t_0)$ to $x(t_0 - h)$; in general, $h$ can be non-uniform across different timesteps. An example of the second-order SRK (SRK2) method is called the Runge–Kutta–Gegenbauer (RKG2) method proposed by Skaras et al. (2021), which takes the form of

$$
\begin{aligned}
\boldsymbol{Y}_0 &= \boldsymbol{x}(t_0), \boldsymbol{Y}_1 = \boldsymbol{Y}_0 - h\tilde{\mu}_1 \boldsymbol{v}(\boldsymbol{Y_0}, t_0), \\
\boldsymbol{Y}_j &= \mu_j \boldsymbol{Y}_{j-1} + \nu_j \boldsymbol{Y}_{j-2} + (1 - \mu_j - \nu_j)\boldsymbol{Y}_0 \\
&\quad - \tilde{\mu}_j h \boldsymbol{v}(\boldsymbol{Y}_{j-1}, t_{j-1}) - \tilde{\gamma}_j h \boldsymbol{v}(\boldsymbol{Y}_0, t_0), \quad j = 2, ..., s, \\
\boldsymbol{x}(t_0 - h) &= \boldsymbol{Y}_s.
\end{aligned}
\tag{3}
$$

Here, the SRK2 coefficients satisfy

$$\mu_j = \frac{2j+1}{j}\frac{b_j}{b_{j-1}}, \tilde{\mu}_j = \mu_j w_1, \nu_j = -\frac{j+1}{j}\frac{b_j}{b_{j-2}}, \tilde{\gamma}_j = -\tilde{\mu}_j a_{j-1};$$

where

$$w_1 = \frac{6}{(s+1)(s-1)}, b_j = \frac{4(j-1)(j+4)}{3j(j+1)(j+2)(j+3)}, a_j = 1 - \frac{(j+1)(j+2)}{2}b_j.$$

---

**Algorithm 1** STORK-4 (Fourth order STORK)

---

**Require:** initial value $\boldsymbol{x}_T$, timesteps $\{\tilde{t}_i\}_{i=1}^M$, velocity network $\boldsymbol{v}(\cdot, \cdot)$, intermediate step number $s$, Taylor order $n = 1, 2, 3$.
$h_i = \tilde{t}_{i-1} - \tilde{t}_i$,
$\boldsymbol{x}(\tilde{t}_{i-1}) = \boldsymbol{x}(\tilde{t}_i) + h_i \boldsymbol{v}(\boldsymbol{x}(\tilde{t}_i), \tilde{t}_i)$
**for** $i = M - 1$ to $M - n$ **do**
$\quad \boldsymbol{x}(\tilde{t}_{i-1}) = \boldsymbol{x}(\tilde{t}_i) + 1.5 h_i \boldsymbol{v}(\boldsymbol{x}(\tilde{t}_i), \tilde{t}_i) - 0.5 h_{i-1} \boldsymbol{v}(\boldsymbol{x}(\tilde{t}_{i-1}), \tilde{t}_{i-1})$
**end for**
**for** $i = M - n - 1$ to $0$ **do**
$\quad \boldsymbol{Y}_0 = \boldsymbol{x}(\tilde{t}_i), \boldsymbol{Y}_1 = \boldsymbol{x}(\tilde{t}_i) + h_i \mu_1 \boldsymbol{v}(\boldsymbol{x}(\tilde{t}_i), \tilde{t}_i)$,
$\quad$ **for** $j = 2$ to $s$ **do**
$\quad\quad \boldsymbol{v}_{\text{approx}}(\boldsymbol{Y}_{j-1}, t_{j-1}) = \text{TaylorExpansion}(n, Y_{j-1}, t_{j-1}, \boldsymbol{x}(\tilde{t}_i), \tilde{t}_i)$,
$\quad\quad$ **if** $j \leq s - 4$ **then**
$\quad\quad\quad \boldsymbol{Y}_j = h_i \mu_j \boldsymbol{v}_{\text{approx}}(\boldsymbol{Y}_{j-1}, t_{j-1}) - \nu_j \boldsymbol{Y}_{j-1} - \kappa_j \boldsymbol{Y}_{j-2}$,
$\quad\quad$ **else**
$\quad\quad\quad \boldsymbol{Y}_j = \boldsymbol{Y}_{s-4} + h_i \mu_j \boldsymbol{v}_{\text{approx}}(\boldsymbol{Y}_{j-1}, t_{j-1})$,
$\quad\quad$ **end if**
$\quad$ **end for**
$\quad \boldsymbol{x}(\tilde{t}_{i-1}) = \boldsymbol{Y}_s$.
**end for**
**return** $\boldsymbol{x}(\tilde{t}_0)$

---

A fourth-order SRK method (SRK4) is the orthogonal Runge–Kutta-Chebyshev method developed by Abdulle (2002). The $s$-stage SRK4 method for solving this ODE from $x(t_0)$ to $x(t_0 - h)$ is

$$
\begin{aligned}
& \boldsymbol{Y}_0 = \boldsymbol{x}(t_0), \boldsymbol{Y}_1 = \boldsymbol{Y}_0 - h \mu_1 \boldsymbol{v}(\boldsymbol{Y}_0, t_0), \\
& \boldsymbol{Y}_j = -h \mu_j \boldsymbol{v}(\boldsymbol{Y}_{j-1}, t_{j-1}) - \nu_j \boldsymbol{Y}_{j-1} - \kappa_j \boldsymbol{Y}_{j-2}, \quad j = 2, ..., s - 4, \\
& \boldsymbol{Y}_j = \boldsymbol{Y}_{s-4} - h \mu_j \boldsymbol{v}(\boldsymbol{Y}_{j-1}, t_{j-1}), \quad j = s - 3, ..., s \\
& \boldsymbol{x}(t_0 - h) = \boldsymbol{Y}_s.
\end{aligned}
\tag{4}
$$

All parameters $\mu_j, \nu_j, \kappa_j$ are ODE-independent, pre-computed constants; the intermediate times $t_j \in [t_0 - h, t_0]$ are only dependent on $t_0$ and $t_0 - h$. The exact values of the parameters and more details of the derivation can be found in Abdulle (2002) and Appendix C. The SRK4 method allows one to choose $h \sim \mathcal{O}(s^2)$ maintaining stability (Abdulle, 2002), therefore handles the stiffness. We call an update from $\boldsymbol{Y}_{j-1}$ to $\boldsymbol{Y}_j$ as one *sub-step*, and an update from $\boldsymbol{x}(t_0)$ to $\boldsymbol{x}(t_0 - h)$ as one *super-step*. Notice that in all the derivations above, we never assume any special structures of $\boldsymbol{v}(\boldsymbol{x}(t), t)$. With the stiffness analysis in Appendix C, SRK methods uniquely belongs to both stiff solvers and structure-independent solvers, as shown in Figure 4.

**SRK vs. RK methods.** Classical Runge–Kutta (RK) methods and SRK methods crucially differ in their ability to handle stiffness: RK methods are not stiff equation solvers, while the SRK methods are designed primarily for stiff equations. With more sub-steps $s$, RK methods converge with higher order, while SRK methods handle stiffness better. Naturally, SRK methods can tune $s$ to arbitrarily large number with a general formula, while RK methods cannot. Detailed comparisons can be found in Appendix D.

### 3.2 STORK: SRK WITH VIRTUAL NFE

Despite the success of SRK methods in classical numerical analysis, a fatal issue when applying them on diffusion and flow-matching model sampling is that an $s$-stage SRK method requires $s$ NFEs for updating 1 super-step. In practice, $s$ usually needs to be chosen to be around 10 to 50, leading to an inordinate NFE count. As shown in Table 1, naive application of SRK4 to the CIFAR-10 (Krizhevsky et al., 2009) dataset results in very poor sampling results, especially in the range of small NFEs. Therefore, a mechanism to reduce the NFE count must be applied for the SRK methods to become practical.

A natural choice to reduce NFEs in each super-step is to approximate $\boldsymbol{v}(\boldsymbol{Y}_j(t_j), t_j)$ using the Taylor expansion in the time variable $t$ at $\boldsymbol{v}(\boldsymbol{Y}_0, t_0)$. By treating the velocity as a purely $t$-dependent function,

Table 1: Ablation study comparison to the fourth-order Runge–Kutta method and vanilla fourth-order stabilized Runge–Kutta, on CIFAR-10 Krizhevsky et al. (2009) dataset. The NFEs in parentheses indicate the NFEs used for the RK4 method, since they must be multiples of 4 for RK4.

| Method \ NFE | 10(12) | 20 | 30(32) | 40 | 50(52) |
|---|---|---|---|---|---|
| RK4 | 121.411 | 33.662 | 4.504 | 5.059 | 5.091 |
| SRK4 | 443.812 | 40.828 | 6.225 | 6.324 | 6.167 |
| STORK-4 (Ours) | **5.497** | **4.167** | **3.888** | **3.809** | **3.789** |

Taylor expansion yields

$$\boldsymbol{v}(\boldsymbol{Y}_j(t_j), t_j) = \boldsymbol{v}(\boldsymbol{Y_0}, t_0) + (t_j - t_0)\boldsymbol{v}'(\boldsymbol{Y_0}, t_0) + \frac{(t_j - t_0)^2}{2}\boldsymbol{v}''(\boldsymbol{Y_0}, t_0)$$
$$+ \frac{(t_j - t_0)^3}{6}\boldsymbol{v}'''(\boldsymbol{Y_0}, t_0) + \mathcal{O}((t_j - t_0)^4).$$

Since the exact evaluation of the higher-order derivatives also incurs costs no less than an NFE, we further approximate the derivatives of the velocity field $\boldsymbol{v}(\boldsymbol{Y_0}, t_0)$ using a finite-difference approximation. Except for the several initial super-steps (depending on the order of Taylor expansion used), we store the previously computed velocities and use the forward finite-difference method to get an approximation for the derivatives of the noise or velocity up to the desired order. Regarding the initial steps, we use one Euler's method step followed by 2-step Adams-Bashforth method steps until there are enough points for Taylor expansion approximation. The intermediate $\boldsymbol{v}_{\text{approx}}(\boldsymbol{Y}_j, t_j)$ are called *virtual* NFEs, since they are approximated using the Taylor expansion so that no additional NFEs are used for those points as opposed to the actual NFEs.

Plugging in the Taylor expansion and the velocity derivative approximations above into SRK4 described by equation 4 with suitable modification, we get the fourth-order *Stabilized Taylor Orthogonal Runge-Kutta (STORK)* method, denoted by STORK-4. The overall STORK method pipeline is illustrated in Figure 3. Similarly, we denote the first and second order STORK methods as STORK-1 and STORK-2. A similar derivation works for noise-based DMs by using Taylor expansion on the noise. Algorithm 1 summarizes the complete STORK-4 algorithm, with

TaylorExpansion(order, $\boldsymbol{Y}_j(t_j), t_j, \boldsymbol{Y}_0, t_0$)

$$= \begin{cases} \boldsymbol{v}(\boldsymbol{Y}_0, t_0) + (t_j - t_0)\boldsymbol{v}'_{\text{approx}}(\boldsymbol{Y}_0, t_0), & \text{order} = 1. \\ \boldsymbol{v}(\boldsymbol{Y}_0, t_0) + (t_j - t_0)\boldsymbol{v}'_{\text{approx}}(\boldsymbol{Y}_0, t_0) + \frac{(t_j - t_0)^2}{2}\boldsymbol{v}''_{\text{approx}}(\boldsymbol{Y}_0, t_0), & \text{order} = 2. \\ \boldsymbol{v}(\boldsymbol{Y}_0, t_0) + (t_j - t_0)\boldsymbol{v}'_{\text{approx}}(\boldsymbol{Y}_0, t_0) + \frac{(t_j - t_0)^2}{2}\boldsymbol{v}''_{\text{approx}}(\boldsymbol{Y}_0, t_0) & \\ \quad + \frac{(t_j - t_0)^3}{6}\boldsymbol{v}'''_{\text{approx}}(\boldsymbol{Y}_0, t_0), & \text{order} = 3. \end{cases}$$

More the details of the algorithms are in the Appendix E.

We note that only first, second, and fourth order SRK methods exist for reasons related to the roots of the so-called stability polynomial (Abdulle, 2002). Therefore, third and higher-order SRK methods do not exist, and STORK-$k$ can exist only for $k = 1, 2, 4$. We find that STORK-4 consistently outperforms STORK-1 and STORK-2, so we conduct all the experiments using the STORK-4 method. The Taylor expansion order $n$ is empirically chosen to be $n = 2$ with both unconditional and conditional noise-based generation, and $n = 1$ with conditional latent space flow-matching generation. Even though the choice of $s$ does not affect the NFEs in the STORK method, the number of sub-steps $s$ also requires tuning to maximize image quality. Intuitively, excessively large $s$ causes errors of the Taylor expansion to accumulate; excessively small $s$ does not provide a large enough region of absolute stability (Burden & Faires, 2011). More studies on order and $s$ are in Appendix G.1.

Finally, via classical numerical analysis arguments in Abdulle (2002); Meyer et al. (2014); Skaras et al. (2021) and the error term in the Taylor expansion, the order of convergence can be summarized as shown in the following theorem and proved in Appendix F.

**Theorem 1.** *Assume $\boldsymbol{\epsilon}_\theta(\boldsymbol{x}_t, t)$ or $\boldsymbol{v}(\boldsymbol{x}_t, t)$ satisfies the assumptions in Appendix. Let $\{\tilde{\boldsymbol{x}}_{t_i}\}_{i=0}^M$ be the sequence computed by STORK-$k$ with timesteps $\{t_i\}_{i=0}^M$. For $k = 2, 4$, if Taylor expansion is not used for the virtual NFEs, the STORK-$k$ solver converges to the expected solution with order $k$,*

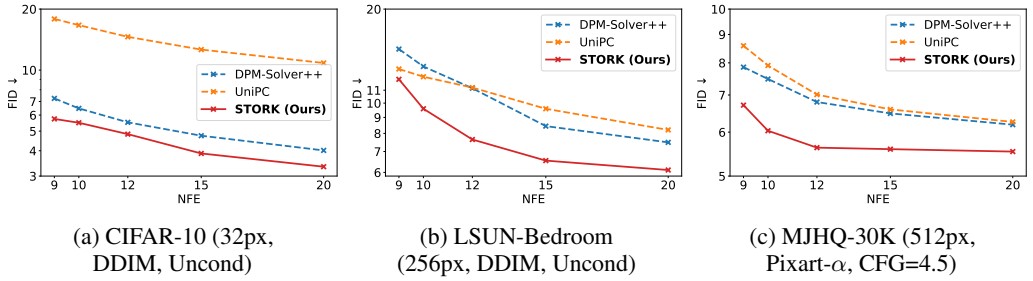

Figure 5: Sample quality measured by FID ↓ for unconditional (Uncond) and classifier-free-guided (CFG) generation with noise-prediction models. As shown, STORK constantly outperforms other methods across datasets and image scales.

*i.e.* $\tilde{\boldsymbol{x}}_{t_0} - \boldsymbol{x}_0 \sim \mathcal{O}(h^k)$ *where* $h := \max_{1 \le i \le M} (t_i - t_{i-1})$. *The Taylor expansion STORK-k solver in Algorithm 2 and 1 converges to the non-Taylor STORK-k solver with order* $\mathcal{O}(h^2)$.

# 4 EXPERIMENTS

We compare the image and video generation quality of STORK against the most state-of-the-art (SOTA) training-free sampling methods, DPM-Solver++ (Lu et al., 2025) and UniPC (Zhao et al., 2023). For image generation, we consider unconditional generation and conditional generation using both pixel and latent-space noise-predicting models (Song et al., 2022; Chen et al., 2023) and latent-space flow-matching models (Xie et al., 2025; Esser et al., 2024; BlackForest, 2024). To demonstrate the broad applicability of STORK, we benchmark on video generation using the Hunyuan Video (Kong et al., 2024) model. We also conduct an inference time and memory consumption analysis across different sampling methods to show that STORK-4 requires similar wall-clock running time and memory compared to other methods when using same amount of NFEs.

Following from Section 3, unless explicitly mentioned, all the noise-based generations use STORK-4 in Algorithm 1 with second-order Taylor expansion, and flow-matching-based generations use STORK-4 with first-order Taylor expansion. Experiment details, additional metrics, and supplementary visualizations are provided in Appendices G and H.

As shown in Figure 5, 6, and Table 2, STORK consistently outperforms SOTA methods **across tasks, model scales, and generation scales**. We believe empirical evidence strongly favors STORK as a better training-free sampling method.

## 4.1 IMAGE GENERATION: UNCONDITIONAL AND CONDITIONAL NOISE-PREDICTING MODELS

**Experimental setup.** We conduct both unconditional and conditional generations using both pixel-space and latent-space models. For unconditional generation, we experiment on CIFAR-10 ($32 \times 32$) (Krizhevsky et al., 2009) and LSUN-Bedroom ($256 \times 256$) (Yu et al., 2016) using the pre-trained DDIM (Ho et al., 2020) checkpoints provided by PNDM (Liu et al., 2022). For each examined method, we generate 50K images to calculate Fréchet Inception Distance (FID) (Heusel et al., 2018) with respect to the Inception activation statistics provided by the PNDM codebase. For conditional generation, we use the Pixart-$\alpha$ on the MJHQ-30K (Li et al., 2024) dataset with 30,000 samples to calculate FID, with classifier-free guidance (CFG) scale 4.5. By empirical evidence, $s = 14$ is used on the CIFAR-10 dataset, and $s = 24$ is used for LSUN-Bedroom and MJHQ-30K. To avoid the singularity at $t = 0$, we denoise up to a small value $\epsilon > 0$.

**Results.** As shown in Figure 5a, 5b, and 5c, the FID curve for STORK consistently remains at the bottom until all methods converge to a similar value. These results show the robustness of STORK across datasets and NFEs.

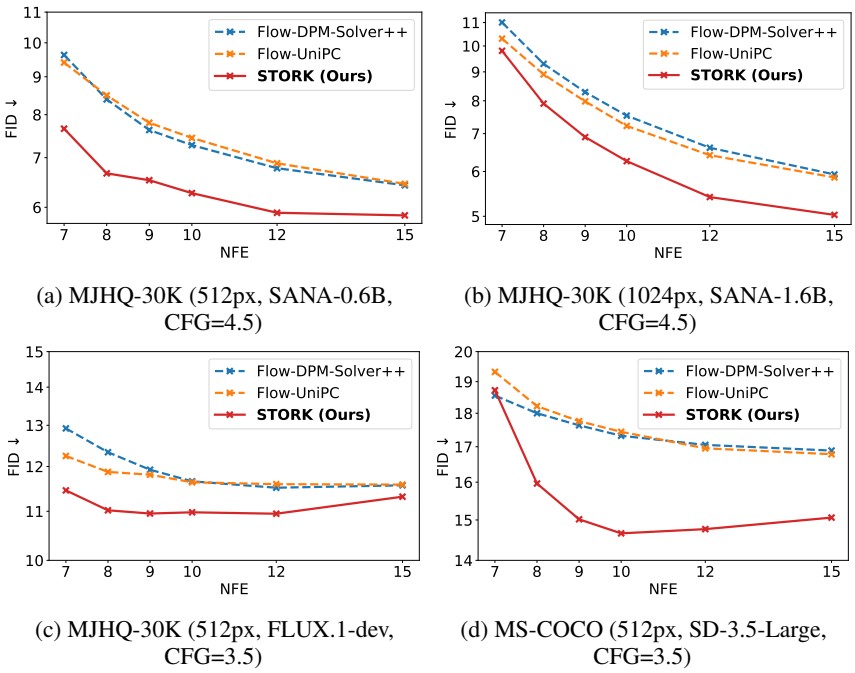

Figure 6: Sample quality measured by FID ↓ for conditional generation with latent space flow-matching models. As shown, STORK constantly outperforms other methods across various text-to-image flow-matching based generative models by a large margin.

## 4.2 IMAGE GENERATION: CONDITIONAL FLOW-MATCHING MODELS

**Experimental setup.** We benchmark STORK on classifier-free text-to-image generation using SANA (Xie et al., 2025), FLUX.1-dev (BlackForest, 2024), and Stable-Diffusion-3.5-Large (SD-3.5-L) (Esser et al., 2024) as our latent-space flow-matching models. To demonstrate the scalability of STORK in terms of model size and generation scale, we use SANA-0.6B to benchmark at 512px resolution and SANA-1.6B to benchmark at 1024px resolution, using prompts from the MJHQ-30K Li et al. (2024) datasets. To further demonstrate STORK's synergy with more popular and larger-scale models, we also benchmark generation at 512px using FLUX.1-dev and SD-3.5-L. Moreover, we prompt SD-3.5-L on the validation split of MS-COCO Lin et al. (2015) dataset to validate the robustness of STORK in terms of data distribution. All reported FIDs are calculated using 30k samples, with reference images resized to the corresponding generation resolution for Inception statistics calculation. Following (Esser et al., 2024), we randomly sampled 30k validation image-prompt pairs from MS-COCO as a reference for FID calculations. Finally, the CFG scales for each benchmark are set to the models' defaults, which are 3.5 for SD-3.5-L and FLUX.1-dev and 4.5 for SANA.

**Results.** As shown in Figure 6a to Figure 6d, STORK demonstrates superior FIDs when using 7-15 NFEs. The non-trivial performance gap from STORK to other sampling methods adds quantitative evidence for the superiority of STORK, besides the qualitative samples from Figure 1.

## 4.3 VIDEO GENERATION: CONDITIONAL LATENT FLOW MATCHING

**Experimental setup.** We finally test our STORK method on the text-to-video generation task. To the best of our knowledge, **we are the first training-free fast sampling work with experiments on video generation**. We use the Hunyuan video diffusion model (Kong et al., 2024), with each frame generated at 512×320 resolution. For each video, 129 frames are generated at 15 frames per second (fps). Classifier-free guidance scale is set to Hunyuan's default value of 6. We benchmark on the EvalCrafter (Liu et al., 2024b) evaluation suite, which consists of 700 prompts. Four different sub-metrics are calculated and aggregated to a final score.

Table 2: 🎥 EvalCrafter ↑ evaluation of Hunyuan video diffusion model, with different sampling methods. Four sub-metrics and the final score are recorded. As shown in the table, STORK method consistently outperforms the Flow-DPM-Solver++ and the Flow-UniPC methods in the final score.

| Method \ NFE | | 4 | 5 | 6 | 7 | 8 |
|---|---|---|---|---|---|---|
| Visual Quality | Flow-DPM-Solver++ | 45.02 | 46.42 | 48.03 | 49.51 | 50.74 |
| | Flow-UniPC | 45.51 | 47.46 | 49.23 | 50.72 | 51.88 |
| | STORK (Ours) | 50.00 | 51.91 | 52.11 | 52.72 | 52.68 |
| Text-Video Alignment | Flow-DPM-Solver++ | 40.90 | 46.10 | 47.54 | 48.78 | 47.83 |
| | Flow-UniPC | 41.38 | 46.35 | 47.43 | 48.60 | 46.59 |
| | STORK (Ours) | 43.18 | 46.11 | 46.64 | 50.09 | 46.92 |
| Motion Quality | Flow-DPM-Solver++ | 55.35 | 54.49 | 54.26 | 53.74 | 53.89 |
| | Flow-UniPC | 55.24 | 54.56 | 54.06 | 54.12 | 53.37 |
| | STORK (Ours) | 55.02 | 54.50 | 54.26 | 54.18 | 54.06 |
| Temporal Consistency | Flow-DPM-Solver++ | 64.17 | 63.80 | 63.48 | 63.25 | 63.07 |
| | Flow-UniPC | 63.94 | 63.43 | 63.20 | 62.92 | 62.76 |
| | STORK (Ours) | 61.41 | 61.67 | 62.08 | 62.14 | 62.26 |
| Final Score | Flow-DPM-Solver++ | 205 | 211 | 213 | 215 | 216 |
| | Flow-UniPC | 206 | 212 | 214 | 216 | 215 |
| | STORK (Ours) | **210** | **214** | **215** | **219** | **216** |

Table 3: Inference wall-clock profiling across different flow-matching sampling methods, all with 10 NFEs, over 10 trials.

| Method | Avg (s) | Min (s) | Max (s) |
|---|---|---|---|
| Flow-DPM-Solver++ | 1.100 | 1.042 | 1.583 |
| Flow-UniPC | 1.223 | 1.167 | 1.699 |
| STORK-4 (Ours) | 1.224 | 1.167 | 1.687 |

Table 4: GPU memory usage during inference for different flow-matching sampling methods, all with 10 NFEs.

| Method | Max alloc. (GB) | Max reserv. (GB) |
|---|---|---|
| Flow-DPM-Solver++ | 32.069 | 32.580 |
| Flow-UniPC | 32.069 | 32.580 |
| STORK-4 (Ours) | 32.071 | 32.584 |

**Results.** Table 2 showcases the text-to-video generation results. As shown in the table, STORK constantly outperforms the flow-DPM-Solver++ (Lu et al., 2025; Xie et al., 2025) and the flow-UniPC (Zhao et al., 2023) methods in terms of the final score. For the sub-metrics, STORK constantly outperforms the other methods in terms of the visual quality, especially when NFEs are extremely small, and achieves comparable results in other metrics. This better video generation further support the general applicability of STORK to various generation tasks.

## 4.4 INFERENCE SPEED AND MEMORY CONSUMPTION ANALYSIS

We finally test the wall-clock time inference and the memory consumptions across different sampling methods. All experiments are conducted on a single H100-NVML GPU and generating images using batch size 1 with FLUX.1-Dev loaded in `torch.bfloat16` generating at (512×512) resolution. For the time efficiency measurement, 10 NFEs are used for each method with 10 trials. As shown in Table 3, the time usage is similar for our proposed STORK-4 compared to other methods. For the memory usage, as shown in Table 4, both the maximum allocated memory and the maximum reserved memory are similar across different methods.

## 5 CONCLUSION

We propose a novel training-free fast sampling method called STORK for unconditional and conditional diffusion and flow matching models. Our method is ODE-structure independent; therefore, it can be applied to noise and flow matching-based DMs without any further adjustments. Nevertheless, sampling using even smaller NFEs with STORK and its possible future extensions remains to be explored. Therefore, further investigation into the potentials of STORK for effective sampling of diffusion and flow matching models presents a promising direction for future research.

ACKNOWLEDGMENTS

Zheng Tan and Andrea L. Bertozzi are supported by NSF grant DMS-2152717, ONR grant DOD N00014-23-1-2565, and DOE grant DE-SC0025589. Ernest K. Ryu is supported by NSF grant CCF-2504627.

The first author would like to thank Sicheng Mo for valuable discussions on diffusion models.

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

APPENDIX

## A  LARGE LANGUAGE MODELS (LLMS) USAGE

We did not use LLMs for the writing of this paper, and LLMs did not help to the extent that they could be regarded as a contributor during the research process.

## B  BRIEF ILLUSTRATION OF STIFFNESS

In this section, we briefly illustrate the classical notion of *stiffness* in numerical analysis. Consider the ODE,

$$\frac{dx}{dt} = -20x, \quad x(0) = 1, \quad t \in [0, 1]. \tag{5}$$

It can be easily shown that $x(t) = e^{-20t}$. However, when the ODE above is solved numerically using various methods, some methods exhibit spurious oscillations, as illustrated in Figure 7. While the numerical solution converges to the exact solution in the limit as the step size $h \to 0$, significant errors can arise for moderate values of $h$, particularly when the analytical solution trajectory is "not straight enough."

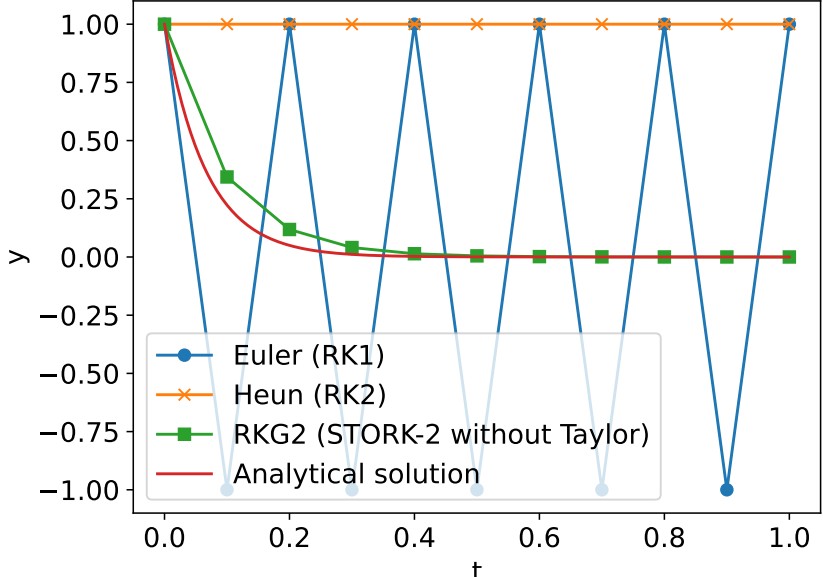

Figure 7: Example of a stiff ODE. The plot shows the analytical and numerical solutions of the ODE equation 5, using Euler, Heun, and the second-order Runge–Kutta–Gegenbauer (RKG2) method with 4 sub-steps. Euler's method and Heun's method are two of the most popular flow matching sampling methods, implemented in diffuser (von Platen et al., 2022). The RKG2 method is our STORK-2 method with exact sub-steps. Ten timesteps are used for each of the numerical solutions. As shown in the plot, Euler and Heun's method has significant errors while RKG2 is close to the exact solution.

The *region of absolute stability* characterizes the notion of stability for numerical methods. Consider the test problem

$$\frac{dx}{dt} = \lambda x, \tag{6}$$

where $\lambda \in \mathbb{C}$ can be an arbitrary complex number. The region of absolute stability of a numerical method is defined as the set of $h\lambda$ values such that the numerical method with step size $h$ applied to equation 6 yields a solution that remains bounded. Since the theoretical exact solution to equation 6

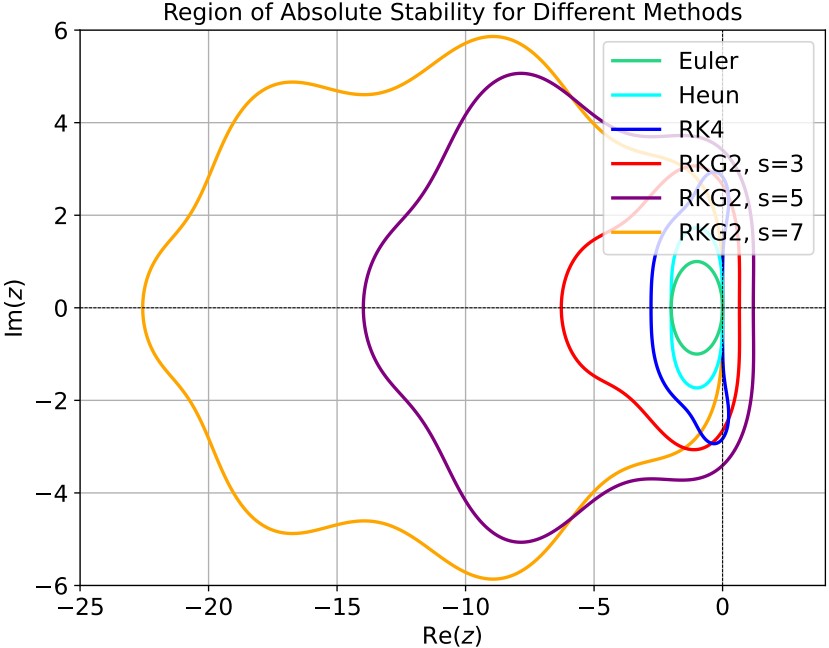

Figure 8: Region of absolute stability for different numerical methods. Euler and Heun's methods are widely used in the diffusion model sampling, especially in the flow matching setting. The RK4 method is the most widely used explicit Runge–Kutta method in numerical analysis. The RKG2 method is the basis for our STORK-2 method; in this plot, 3, 5, and 7 sub-steps are used. As shown in the plot, the RKG2 method has a much larger region of absolute stability than the other methods, so that it is much more stable, therefore more suitable for solving stiff problems. Moreover, the region of absolute stability becomes significantly larger as $s$ increases.

remains bounded for all $\lambda$ in the left half of the complex plane, a numerical method is considered more stable if its region of absolute stability encompasses a larger portion of that half-plane.

As an example, when applying Euler's method to the test problem equation 6, the numerical method becomes

$$x(t+h) = x(t) + h\lambda x(t) = (1+h\lambda)x(t).$$

Therefore, the stability polynomial is $R(z) = 1 + z$; the region of absolute stability is therefore $|1+z| < 1$, which corresponds to a circle centered around 1 with radius 1 in the complex plane.

The region of absolute stability for Heun's method, fourth-order Runge–Kutta (RK4) method, and the second-order Runge–Kutta–Gegenbauer (RKG2) with 4 sub-steps are plotted in Figure 8. The RKG2 method, for which the STORK-2 method is derived based on, has a region of absolute stability that contains a much larger portion of the left half complex plane than the Euler, Heun, and RK4 methods. Analytical derivation in Appendix C further confirms that an $s$-stage RKG2 method has a large region of absolute stability that grows with order $\mathcal{O}(s^2)$.

## C   STABILIZED RUNGE–KUTTA METHODS

We present the derivation of the second-order Runge–Kutta–Gegenbauer (RKG2) method (Skaras et al., 2021) and the fourth order orthogonal Runge–Kutta–Chebyshev (ROCK4) method (Abdulle, 2002). The derivations closely follow the original derivations in Skaras et al. (2021) and Abdulle (2002). Notice RKG2 corresponds to the SRK2 method and ROCK4 corresponds to the SRK4 method in the main content.

Consider the ODE

$$\frac{d\boldsymbol{x}}{dt} = M\boldsymbol{x}, \boldsymbol{x}(0) = \boldsymbol{x}_0; \quad M \in \mathbb{R}^{n\times n}, \boldsymbol{x} \in \mathbb{R}^n, n \in \mathbb{N}_+.$$

The analytical solution for the above linear equation is defined by

$$\boldsymbol{x}(t) = e^{tM}\boldsymbol{x}(0) = \sum_{m=0}^{\infty} \frac{(tM)^m}{m!}\boldsymbol{x}_0. \tag{7}$$

Since we would like to derive a single-step method, the method takes the form of

$$\boldsymbol{x}(t+h) = R(hM)\boldsymbol{x}(t), \tag{8}$$

where $h$ is the timestep size, and $R(hM)$ is called the *stability polynomial*. In order to enlarge the size of the region of absolute stability, the shifted Gegenbauer polynomial with parameter $\alpha = \frac{3}{2}$ is chosen to be the stability polynomial; this leads to the Runge–Kutta–Gegenbauer (RKG) method (Skaras et al., 2021; O'Sullivan, 2019). More precisely, the stability polynomial of an $s$-stage RKG method $R_s(hM)$ satisfies

$$R_s(z) = a_s + b_s C_s^{(3/2)}(1 + w_1 z),$$

where $a_s, b_s, w_1$ are parameters to be chosen, and $C_s^{(3/2)}(x)$ is the $s$-degree Gegenbauer polynomial with parameter $\frac{3}{2}$ (Skaras et al., 2021). In order for the method to be convergent, we need to equate equation 7 and equation 8 to the highest order possible. Since there are three unknowns, the best possible order is to equate the first three terms and get second-order $\mathcal{O}(h^2)$ convergence. Therefore, the three equations that are needed to be satisfied are

$$R_s(0) = R_s'(0) = R_s''(0) = 1.$$

Equating and solving the differential equation, we get the result for RKG2 method:

$$w_1 = \frac{6}{(s+4)(s-1)}, \quad b_j = \frac{4(j-1)(j+4)}{3j(j+1)(j+2)(j+3)}, \quad a_j = 1 - \frac{(j+1)(j+2)}{2}b_j.$$

To ensure numerical stability in Appendix B, one only needs to ensure that $1 + w_1 hM$ has eigenvalue within $[-1, 1]$, so that by the boundedness of the Gegenbauer polynomial we get numerical stability for free (Skaras et al., 2021). This requires that $h = h_{\text{explicit}} \frac{(s+4)(s-1)}{6} \sim \mathcal{O}(s^2)$, where $h_{\text{explicit}}$ is the maximum timesteps that ensures Euler's method to be numerically stable.

Finally, we would like to turn the RKG2 method into the Runge–Kutta form as shown in equation 3, so that it can be implemented easily in practice. Using the well-known Gegenbauer polynomial inductive relationship

$$C_s^{(\alpha)}(z) = \frac{1}{s}\Big[2z(s+\alpha-1)C_{s-1}^{(\alpha)}(z) - (s+2\alpha-2)C_{s-2}^{(\alpha)}(z)\Big],$$

it can be derived that the stability polynomial satisfies the relationship

$$a_j + b_j C_j^{(3/2)}(1 + w_1 z) = \mu_j(a_{j-1} + b_{j-1}C_{j-1}^{(3/2)}(1 + w_1 z)) + \nu_j(a_{j-2} + b_{j-2}C_{j-2}^{(3/2)}(1 + w_1 z))$$
$$+ \tilde{\mu}_j(a_{j-1} + b_{j-1}C_{j-1}^{(3/2)}(1 + w_1 z)) + (1 - \mu_j - \nu_j) + \tilde{\gamma}_j,$$

where

$$\mu_j = \frac{2j+1}{j}\frac{b_j}{b_{j-1}}, \quad \tilde{\mu}_j = \mu_j w_1, \quad \nu_j = -\frac{j+1}{j}\frac{b_j}{b_{j-2}}, \quad \tilde{\gamma}_j = -\tilde{\mu}_j a_{j-1}.$$

Therefore, the RKG2 method in the Runge–Kutta formulation is

$$\begin{aligned}
\boldsymbol{Y}_0 &= \boldsymbol{x}(t), \\
\boldsymbol{Y}_1 &= \boldsymbol{Y}_0 - \tilde{\mu}_1 hM\boldsymbol{Y}_0, \\
\boldsymbol{Y}_1 &= \mu_j\boldsymbol{Y}_{j-1} + \nu_j\boldsymbol{Y}_{j-2} + (1 - \mu_j - \nu_j)\boldsymbol{Y}_0 - \tilde{\mu}_j hM\boldsymbol{Y}_{j-1} - \tilde{\gamma}_j hM\boldsymbol{Y}_0, \quad 2 \le j \le s, \\
\boldsymbol{x}(t-h) &= \boldsymbol{Y}_s.
\end{aligned} \tag{9}$$

Replace each $M\boldsymbol{Y}_j$ in equation 9 by $\boldsymbol{v}(\boldsymbol{Y}_j, t_j)$, we exactly recovers equation 3. Then the derivation of our STORK-2 method follows as in Section 3.

The fourth-order orthogonal Runge–Kutta–Chebyshev (ROCK4) method can be derived similarly, with an extra composition method term. The stability polynomial is designed to be

$$R_s(z) = w_4(z)P_{s-4}(z),$$

which satisfies $|R_s(z)| \leq 1$ for $z \in [-l_s, 0]$, with $l_s$ as large as possible. The first polynomial

$$w_4(z) = \left(1 - \frac{1}{z_1}\right)\left(1 - \frac{1}{\overline{z}_1}\right)\left(1 - \frac{1}{z_2}\right)\left(1 - \frac{1}{\overline{z}_2}\right)$$

is a fourth order polynomial that serves as a composition method (Burden & Faires, 2011) to boost the entire method to fourth order, and the second polynomial $P_{s-4}(z)$ is an orthogonal polynomial that is orthogonal with respect to the weight function $\frac{w_4(z)^2}{\sqrt{1-z^2}}$ and normalized such that $P_{s-4}(0) = 1$.

Detailed derivation of the polynomial $P_{s-4}(z)$ is similar to the RKG2 method and can be found in Abdulle (2002). For the expression of $w_4(z)$ that serves as a composition method, it evokes the concept of *Butcher's Tableau* (Burden & Faires, 2011) in classical numerical analysis, and the coefficients are derived by solving a system of $8$ equations with $10$ unknown variables, so that two degrees of freedom exist and the original ROCK4 paper Abdulle (2002) chose the optimal parameters by experiments. The coefficients are precomputed and provided along with the supplementary material. Both our implementation and the coefficients in the STORK-4 method are heavily based on the implementation in Steinebach (2006).

## D  COMPARISON WITH SIMILAR METHODS

In this section, we compare STORK with existing sampling methods and highlight their relationships. The following discussion motivates the effectiveness of STORK, and the ablation study in Table 1 shows the necessity of modifications from stabilized Runge–Kutta to STORK methods.

### D.1  COMPARISON BETWEEN STORK AND CLASSICAL RUNGE–KUTTA METHODS

When $s = 1$, STORK-2 and STORK-4 both reduce to Euler's Method, which is a 1-step Runge–Kutta (RK) method. However, for $s > 1$, STORK-2 and STORK-4 are not classical RK methods, and they exhibit some fundamental qualitative differences. The classical RK methods, such as Heun's method or RK4, are designed for higher-order accuracy as the number of sub-steps increases (Burden & Faires, 2011), while the STORK methods (or SRK methods) are designed to address the stiffness in ODEs and PDEs (Verwer et al., 1990; Abdulle, 2002) with regard to the step size. Regardless of the choice of sub-step number $s$, the STORK-$k$ method in Algorithm 2 and 1 with a fixed $k$ always has the same order as shown in Theorem 1.

Another essential difference is that the number of sub-steps $s$ in STORK is an easily tunable hyperparameter. For stiffer problems, one can straightforwardly increase $s$. With $s$-step RK methods, the parameter $s$ is usually not considered a hyperparameter, since changing $s$ completely changes the method. By tuning $s$ and using many sub-steps between the super-step from $t_i$ to $t_{i-1}$, STORK can effectively address stiffness, while direct applications of traditional RK methods have not lead to significant improvements in the sampling quality (Liu et al., 2022; Lu et al., 2022) due to the stepsize constraint that stiffness imposes on RK methods.

### D.2  COMPARISON BETWEEN STORK AND VANILLA STABILIZED RUNGE–KUTTA METHODS

Although the SRK method is a single-step method, *i.e.*, one only needs the value of $\boldsymbol{x}(t)$ in order to get the value of $\boldsymbol{x}(t - h)$, the Taylor expansion makes STORK a multi-step method (Burden & Faires, 2011). However, STORK is not a *linear* multi-step method. A linear multi-step method, such as the 2-step Adams–Bashforth method

$$\boldsymbol{x}(t - h) = \boldsymbol{x}(t) - \frac{h}{2}[f(\boldsymbol{x}(t), t) + f(\boldsymbol{x}(t + h), t + h)]$$

reuses each previous function evaluation once per step. Other pseudo multi-step numerical methods, such as PNDM (Liu et al., 2022), also have the same mechanism. STORK, on the other hand, repeatedly uses previous function evaluations in the Taylor expansion approximation. Intuitively speaking, more information about the local velocity fields in between $t$ and $t - h$ is extracted from those evaluations, so that the solution would follow the learned velocity field more closely, leading to a better approximation. Experiments in Section 4 demonstrate the effectiveness of this mechanism.

---

**Algorithm 2** STORK-2 (Second order STORK)

---

**Require:** initial value $\boldsymbol{x}_T$, timesteps $\{\tilde{t}_i\}_{i=1}^M$, velocity network $\boldsymbol{v}(\cdot, \cdot)$, intermediate step number $s$, Taylor order $n = 1, 2, 3$.
$h_i = \tilde{t}_{i-1} - \tilde{t}_i$,
$\boldsymbol{x}(\tilde{t}_{i-1}) = x(\tilde{t}_i) + h_i \boldsymbol{v}(x(\tilde{t}_i), \tilde{t}_i)$
**for** $i = M - 1$ to $M - n$ **do**
  $\boldsymbol{x}(\tilde{t}_{i-1}) = x(\tilde{t}_i) + 1.5 h_i \boldsymbol{v}(x(\tilde{t}_i), \tilde{t}_i) - 0.5 h_{i-1} \boldsymbol{v}(x(\tilde{t}_{i-1}), \tilde{t}_{i-1})$
**end for**
**for** $i = M - n - 1$ to $0$ **do**
  $\boldsymbol{Y}_0 = \boldsymbol{x}(\tilde{t}_i)$, $\boldsymbol{Y}_1 = \boldsymbol{x}(\tilde{t}_i) + h_i \mu_1 \boldsymbol{v}(\boldsymbol{x}(\tilde{t}_i), \tilde{t}_i)$,
  **for** $j = 2$ to $s$ **do**
    $\boldsymbol{v}_{\text{approx}}(\boldsymbol{Y}_{j-1}, t_{j-1}) = \text{TaylorExpansion}(n, Y_{j-1}, t_{j-1}, \boldsymbol{Y}_0, \tilde{t}_i)$,
    $\boldsymbol{Y}_j = \mu_j \boldsymbol{Y}_{j-1} + \nu_j \boldsymbol{Y}_{j-2} + (1 - \mu_j - \nu_j) \boldsymbol{Y}_0 + \tilde{\mu}_j h_i \boldsymbol{v}_{\text{approx}}(\boldsymbol{Y}_{j-1}, t_{j-1}) + \tilde{\gamma}_j h_i \boldsymbol{v}(\boldsymbol{Y}_0, \tilde{t}_i)$,
  **end for**
  $\boldsymbol{x}(\tilde{t}_{i-1}) = \boldsymbol{Y}_s$.
**end for**
**return** $x(\tilde{t}_0)$

---

# E   STORK-2 AND STORK-4 ALGORITHMS

We now present the details of STORK-2 and STORK-4 algorithms described in Section 3. For both methods, one Euler's step is used at the beginning. If Taylor expansion order is 1, then one more step of 2-step Adams-Bashforth method is used, and if Taylor expansion order is 2, two more step of 2-step Adams-Bashforth method is used. The later steps follow the derivation of SRK2 and SRK4 methods in Section 3. Details of the algorithms are shown in Algorithm 2 and Algorithm 1 for flow-based models. Similar algorithms can be written down using exactly the same procedure as in Section 3 on noise-based models, by Taylor expansion on the noise. We later noticed that we essentially swapped $\boldsymbol{Y}_0$ and $\boldsymbol{Y}_1$ in our official STORK-4 implementation, so the same change needs to be done if one wants to reproduce the results in the paper.

# F   CONVERGENCE PROOF FOR THEOREM 1

In this section, we present the proof of Theorem 1.

*Proof of Theorem 1.* By derivation in C, the RKG2 method converges with order $\mathcal{O}(h^2)$ and the ROCK4 method converges with order $\mathcal{O}(h^4)$ as expected.

Now we'll show that the STORK-2 method and STORK-4 method converge to RKG2 and ROCK4, respectively, with order $\mathcal{O}(h^2)$ for both cases. By Taylor expansion, we know that

$$\boldsymbol{v}(\boldsymbol{Y}_j(t_j), t_j) = \boldsymbol{v}(\boldsymbol{Y_0}, t_0) + (t_j - t_0)\boldsymbol{v}'(\boldsymbol{Y_0}, t_0) + \mathcal{O}((t_j - t_0)^2).$$

Since the three point forward velocity approximation has order $\mathcal{O}(h)$ by classical numerical analysis, and $t_j - t_0 \leq h$ in both STORK-2 and STORK-4, we know that

$$\boldsymbol{v}(\boldsymbol{Y}_j(t_j), t_j) = \boldsymbol{v}(\boldsymbol{Y_0}, t_0) + (t_j - t_0)\boldsymbol{v}'_{\text{approx}}(\boldsymbol{Y_0}, t_0) + \frac{(t_j - t_0)^2}{2}\boldsymbol{v}''_{\text{approx}}(\boldsymbol{Y_0}, t_0) + \mathcal{O}(h).$$

Plugging into Algorithm 2 and Algorithm 1, we know that the STORK-2 and STORK-4 converge to RKG2 equation 3 and ROCK4 equation 4 respectively, with order $\mathcal{O}(h^2)$ since in both STORK-2 and STORK-4, there exists an additional $h$ in front of each virtual NFE $\boldsymbol{v}(\boldsymbol{Y}_j(t_j), t_j)$. □

# G   EXPERIMENTS

In this section, we discuss our experiment setup in greater detail. We present all available data in tables for clearer demonstration. For UniPC (Zhao et al., 2023), DPM-Solver++ (Lu et al., 2025), DEIS (Zhang & Chen, 2023), Flow-Euler, and DDIM (Song et al., 2022) schedulers, we

use the implementation provided by the `Diffusers` (von Platen et al., 2022) package version `0.35.0.dev0`.

Note that in order for the compatibility of FLUX.1-dev and Hunyuan Video (Kong et al., 2024) pipelines with DPM-Solver++ and UniPC, we made two lines of modification on the original Diffusers source code. Essentially, we enforce `sigmas = None` if the scheduler's config has `use_flow_sigmas = True`. We can provide the updated implementation upon request for reproducibility.

### G.1 STUDIES ON THE PARAMETERS

In this subsection, we conduct ablation studies on the number of sub-steps $s$ used, the order of the STORK method, and the order of the Taylor expansion.

#### G.1.1 EFFECT OF SUB-STEPS

We investigate the effect of sub-steps on the generation fidelity of diffusion models. For these experiments, we use the SANA (Xie et al., 2025) 0.6B variant and generate images at $512 \times 512$ resolution for FID calculation using 30000 samples. As shown in Table 5, while all choices of $s$ lead to decent results, the choice of $s$ cannot be excessively large or small. However, larger $s$ enables better stiffness handling ability, overly large $s$ results in too much Taylor series approximation errors. We found that $s = 9$ is optimal for the conditional flow-matching generation, and recommend users to experiment with choices of $s \leq 100$ for the downstream models and tasks.

Table 5: Effects of sub-steps on sample fidelity. As shown, sub-step $s$ leads to decent performance, and we empirically find $s = 9$ achieves the best performance with SANA-0.6B by (Xie et al., 2025) generating at $512 \times 512$ resolution and in general for flow-matching models.

| $s$ \ NFE | 7 | 8 | 9 | 10 |
|---|---|---|---|---|
| $s$=5 | 8.000 | 7.999 | 8.537 | 9.168 |
| $s$=9 | **7.659** | **6.667** | **6.526** | **6.270** |
| $s$=14 | 8.351 | 7.242 | 7.026 | 6.671 |
| $s$=24 | 8.686 | 7.526 | 7.286 | 6.884 |
| $s$=54 | 8.836 | 7.649 | 7.384 | 6.992 |
| $s$=104 | 8.866 | 7.669 | 7.412 | 7.016 |

#### G.1.2 EFFECT OF SOLVER ORDER AND TAYLOR EXPANSION ORDER

We further conducted ablation studies for both Taylor expansion order and solver order. As mentioned in Section 3, STORK-1, STORK-2, and STORK-4 are the only possible configurations for SRK methods. We now examine their corresponding performance using the SANA-0.6B model for $512 \times 512$ resolution on the MJHQ-30K datasets, using $s = 9$ for all trials. First, second, and third order Taylor expansions for each solver order case is tested. As shown in Table 6, the best combination is STORK-4 with first order Taylor expansion.

### G.2 IMAGE GENERATION METRICS

In this section, we provide the detailed benchmarks on Fréchet Inception Distance (FID) as reported in the main paper. We further benchmark human preference of the generated samples using HPSv2 (Wu et al., 2023), and we report the averaged metrics across categories. Unless otherwise specified, the STORK's hyperparameter $s$ is set to $s = 9$.

As shown in this section, STORK generally outperforms other fast sampling methods, from noise-predicting to flow-matching models across image and video generation tasks and datasets. Moreover, STORK demonstrates scalable performance in terms of model size and generation resolution. Therefore, experimental results strongly support the superiority of STORK as a fast sampling method.

Table 6: Effects of order. STORK-1, STORK-2, STORK-3 denote the order of SRK method as the base for STORK, and "1st", "2nd", and "3rd" denote the Taylor expansion order. FID ↓ for MJHQ-30K using SANA 0.6B (Xie et al., 2025) is tested. The gray numbers denote the best result in the current order, and the bold numbers denote the absolute best numbers.

| Method \ NFE | 7 | 8 | 9 | 10 |
|---|---|---|---|---|
| STORK-1-1st | 23.015 | 21.372 | 20.799 | 19.994 |
| STORK-1-2nd | 33.856 | 25.460 | 22.351 | 20.913 |
| STORK-1-3rd | 22.768 | 21.066 | 20.316 | 19.593 |
| STORK-2-1st | 8.485 | 7.585 | 7.417 | 7.052 |
| STORK-2-2nd | 23.602 | 12.646 | 8.561 | 7.227 |
| STORK-2-3rd | 14.420 | 11.790 | 10.479 | 9.442 |
| STORK-4-1st | **7.659** | **6.667** | **6.526** | **6.270** |
| STORK-4-2nd | 54.179 | 30.168 | 15.254 | 9.619 |
| STORK-4-3rd | 20.484 | 14.976 | 12.556 | 10.712 |

### G.2.1 Fréchet Inception Distance

**Unconditional generation**. The benchmark on CIFAR-10 (Krizhevsky et al., 2009) is presented in table 7. The benchmark on LSUN-Bedroom (Yu et al., 2016) is presented in table 8. For these datasets, the FID is calculated using 50,000 samples, and we use the Inception statistics provided by PNDM (Liu et al., 2022) as reference.

**Conditional generation**. The benchmark for MJHQ-30K (Li et al., 2024) dataset using Pixart-$\alpha$ (Chen et al., 2023) is shown in table 9. The benchmark for MJHQ-30K using SANA-0.6B (Xie et al., 2025) is shown in table 10. The benchmark for MJHQ-30K at 1024 resolution using SANA-1.6B is shown in table 11. The benchmark for MJHQ-30K using FLUX.1-dev (BlackForest, 2024) is shown in table 12. The benchmark for MS-COCO (Lin et al., 2015) using Stable-Diffusion-3.5-Large (Esser et al., 2024) is presented in table 13. All FIDs are calculated using 30,000 samples to expedite experiments. For MS-COCO, we randomly subsampled 30,000 images from the entire validation split. Additionally, the weights for Stable-Diffusion-3.5-Large and FLUX.1-dev are loaded in `torch.bfloat16`.

Table 7: Unconditional generation on CIFAR-10 dataset. (DDIM, 32px)

| Method \ NFE | 8 | 9 | 10 | 12 | 15 | 20 | 30 | 50 | 100 |
|---|---|---|---|---|---|---|---|---|---|
| DDIM (Song et al., 2022) | 23.260 | 20.390 | 18.500 | 15.477 | 12.848 | 10.900 | 8.657 | 6.990 | 5.520 |
| DPM-Solver++ (Lu et al., 2025) | 8.669 | 7.250 | 6.471 | 5.525 | 4.745 | 4.015 | 3.947 | 3.859 | 3.851 |
| UniPC (Zhao et al., 2023) | 19.732 | 17.879 | 16.666 | 14.593 | 12.623 | 10.843 | 8.828 | 7.085 | 5.685 |
| STORK-4, $\epsilon$=1e-2, $s$=14 | **6.753** | **5.743** | **5.497** | 4.964 | 4.592 | 4.168 | 3.888 | 3.789 | - |
| STORK-4, $\epsilon$=1e-3, $s$=14 | 7.816 | 7.505 | 6.077 | **4.831** | **3.879** | **3.337** | **3.204** | **3.484** | - |

Table 8: Unconditional generation on the LSUN-Bedroom dataset. (DDIM, 256px)

| Method \ NFE | 8 | 9 | 10 | 12 | 15 | 20 | 30 | 40 | 50 |
|---|---|---|---|---|---|---|---|---|---|
| DDIM | 22.164 | 18.730 | 16.355 | 13.228 | 10.566 | 8.324 | 6.768 | 6.088 | 5.880 |
| DPM-Solver++ | 17.679 | 14.894 | 13.101 | 11.160 | 8.441 | 7.488 | 6.934 | 6.665 | 6.491 |
| UniPC | **13.665** | 12.858 | 12.146 | 11.213 | 9.592 | 8.208 | 6.481 | **6.043** | **5.772** |
| STORK-4, $\epsilon$=1e-2, $s$=24 | 16.353 | **11.918** | **9.594** | **7.647** | **6.546** | **6.107** | **6.202** | 6.457 | 6.658 |

Table 9: Conditional generation on MJHQ-30K. Our method more quickly converges to a plateau value around 5.5. (Pixart-$\alpha$, 512px, CFG=4.5)

| Method | 8 | 9 | 10 | 12 | 15 | 20 | 30 | 40 | 50 |
|---|---|---|---|---|---|---|---|---|---|
| DEIS (Zhang & Chen, 2023) | 9.046 | 8.162 | 7.645 | 6.835 | **5.535** | **5.528** | 5.525 | 5.540 | 5.544 |
| DPM-Solver++ | **8.734** | 7.859 | 7.481 | 6.799 | 6.485 | 6.189 | 5.952 | 5.846 | 5.764 |
| UniPC | 9.361 | 8.588 | 7.912 | 7.012 | 6.594 | 6.260 | 5.923 | 5.813 | 5.738 |
| STORK-4, $\epsilon$=1e-2, $s$=24 | 8.835 | **6.712** | **6.035** | **5.626** | 5.591 | 5.535 | **5.505** | **5.495** | **5.508** |

Table 10: Conditional generation on MJHQ-30K (SANA-0.6B, 512px, CFG=4.5)

| Method | 7 | 8 | 9 | 10 | 12 | 15 | 20 | 30 | 40 | 50 |
|---|---|---|---|---|---|---|---|---|---|---|
| Flow-Euler | 15.856 | 13.320 | 11.851 | 10.782 | 9.434 | 8.430 | 7.551 | 6.932 | 6.637 | 6.494 |
| Flow-DPM-Solver++ | 9.629 | 8.390 | 7.628 | 7.278 | 6.774 | 6.424 | 6.282 | 6.095 | 6.086 | 6.097 |
| Flow-UniPC | 9.406 | 8.493 | 7.801 | 7.443 | 6.878 | 6.452 | 6.265 | 6.112 | 6.090 | 6.085 |
| STORK-4 | **7.659** | **6.667** | **6.526** | **6.270** | **5.899** | **5.851** | **5.897** | **5.928** | **5.989** | **5.995** |

Table 11: Conditional generation on MJHQ-30K (SANA-1.6B, 1024px, CFG=4.5)

| Method | 7 | 8 | 9 | 10 | 12 | 15 | 20 |
|---|---|---|---|---|---|---|---|
| Flow-Euler | 18.348 | 14.849 | 12.603 | 10.980 | 8.828 | 7.142 | 6.069 |
| Flow-DPM-Solver++ | 11.005 | 9.299 | 8.288 | 7.532 | 6.609 | 5.929 | 5.428 |
| Flow-UniPC | 10.304 | 8.909 | 7.981 | 7.223 | 6.412 | 5.856 | 5.448 |
| STORK-4 | **9.804** | **7.910** | **6.901** | **6.258** | **5.407** | **5.029** | **4.879** |

Table 12: Conditional generation on MJHQ-30K (FLUX.1-dev, 512px, CFG=3.5)

| Method | 7 | 8 | 9 | 10 | 12 | 15 | 20 |
|---|---|---|---|---|---|---|---|
| Flow-Euler | 16.046 | 14.609 | 13.968 | 13.493 | 12.804 | 12.588 | 12.467 |
| Flow-DPM-Solver++ | 12.922 | 12.337 | 11.925 | 11.659 | 11.515 | 11.569 | 11.779 |
| Flow-UniPC | 12.248 | 11.874 | 11.810 | 11.637 | 11.595 | 11.585 | 12.044 |
| STORK-4 | **11.456** | **11.021** | **10.952** | **10.978** | **10.948** | **11.316** | **11.705** |

Table 13: Conditional generation on MS-COCO (SD-3.5-Large, 512px, CFG=3.5)

| Method | 7 | 8 | 9 | 10 | 12 | 15 | 20 |
|---|---|---|---|---|---|---|---|
| Flow-Euler | 20.748 | 19.037 | 18.408 | 17.975 | 17.268 | 17.078 | 16.858 |
| Flow-DPM-Solver++ | **18.553** | 18.001 | 17.633 | 17.322 | 17.053 | 16.889 | 16.633 |
| Flow-UniPC | 19.319 | 18.220 | 17.759 | 17.439 | 16.954 | 16.784 | 16.628 |
| STORK-4 | 18.722 | **15.964** | **15.017** | **14.661** | **14.767** | **15.061** | **15.291** |

### G.2.2  HPSV2

We further evaluate STORK in terms of human preferences using HPSv2 (Wu et al., 2023) benchmark, and we report the averaged score over categories in this section. As shown in table 14 and table 15, generated samples by STORK is consistently preferred over other sampling methods.

Table 14: HPSv2 Benchmark. (Pixart-$\alpha$, 512px, CFG=4.5)

| Method \ NFE | 7 | 8 | 9 | 10 | 12 | 15 | 20 | 30 | 40 | 50 |
|---|---|---|---|---|---|---|---|---|---|---|
| DEIS | 27.28 | 28.09 | 28.56 | 28.85 | 29.27 | 29.80 | 29.89 | 29.97 | 29.95 | 29.99 |
| DPM-Solver++ | **27.60** | **28.38** | **28.86** | 29.12 | 29.48 | 29.75 | 29.90 | 29.98 | 29.99 | 30.02 |
| UniPC | 27.33 | 28.17 | 28.66 | 29.04 | 29.51 | 29.78 | 29.94 | 30.02 | 30.04 | 30.05 |
| STORK-4, $\epsilon$=1e-2, $s$=24 | 27.21 | 28.00 | 28.75 | **29.17** | **29.66** | **29.92** | **30.03** | **30.07** | **30.09** | **30.08** |

Table 15: HPSv2 Benchmark. (FLUX.1-dev, 512px, CFG=3.5)

| Method \ NFE | 7 | 8 | 9 | 10 | 12 | 15 | 20 | 30 |
|---|---|---|---|---|---|---|---|---|
| Flow-Euler | 28.60 | 29.09 | 29.52 | 29.69 | 30.13 | 30.33 | 30.48 | 30.64 |
| Flow-DPM-Solver++ | 29.34 | 29.58 | 29.93 | 30.09 | 30.30 | 30.51 | 30.60 | 30.66 |
| Flow-UniPC | 29.61 | 29.77 | 30.07 | 30.21 | 30.39 | 30.56 | 30.60 | 30.67 |
| STORK-4 | **30.27** | **30.63** | **30.84** | **30.84** | **31.00** | **30.95** | **30.90** | **30.71** |

## G.3 Video Generation Metrics 🎥

In this section, we provide more comprehensive results of the text-to-video generation using Hunyuan model (Kong et al., 2024), up to NFE=30. As can be seen, the final score converges for all the methods after 10 NFEs, and STORK generates the best results in terms of the final score. For the sub-metrics, our STORK method wins by majority in visual quality and motion quality, and get comparable results in other metrics.

Table 16: EvalCrafter evaluation of Hunyuan Model, with different sampling methods. Scores of the four sub-metrics and the total score are recorded. As shown in the table, STORK method consistently outperforms the flow-DPM-Solver++ and the flow-UniPC methods in terms of the final score for small NFEs, all get similar final score for larger NFEs.

| | Method \ NFE | 4 | 5 | 6 | 7 | 8 | 9 | 10 | 12 | 20 | 30 |
|---|---|---|---|---|---|---|---|---|---|---|---|
| Visual Quality | Flow-DPM-Solver++ | 45.02 | 46.42 | 48.03 | 49.51 | 50.74 | 51.48 | 52.40 | 53.11 | 54.82 | 54.90 |
| | Flow-UniPC | 45.51 | 47.46 | 49.23 | 50.72 | 51.88 | 52.66 | 53.32 | 54.15 | 55.08 | 55.17 |
| | STORK (Ours) | 50.00 | 51.91 | 52.11 | 52.72 | 52.68 | 53.17 | 53.60 | 53.74 | 53.75 | 53.14 |
| Text-Video Alignment | Flow-DPM-Solver++ | 40.90 | 46.10 | 47.54 | 48.78 | 47.83 | 47.93 | 50.13 | 48.97 | 50.07 | 49.85 |
| | Flow-UniPC | 41.38 | 46.35 | 47.43 | 48.60 | 46.59 | 48.76 | 49.95 | 50.46 | 51.16 | 50.37 |
| | STORK (Ours) | 43.18 | 46.11 | 46.64 | 50.09 | 46.92 | 48.94 | 48.01 | 49.37 | 50.40 | 52.44 |
| Motion Quality | Flow-DPM-Solver++ | 55.35 | 54.49 | 54.26 | 53.74 | 53.89 | 53.55 | 53.50 | 53.54 | 53.22 | 53.62 |
| | Flow-UniPC | 55.24 | 54.56 | 54.06 | 54.12 | 53.37 | 53.51 | 53.40 | 53.43 | 53.32 | 53.63 |
| | STORK (Ours) | 55.02 | 54.50 | 54.26 | 54.18 | 54.06 | 53.47 | 53.79 | 53.81 | 53.95 | 53.80 |
| Temporal Consistency | Flow-DPM-Solver++ | 64.17 | 63.80 | 63.48 | 63.25 | 63.07 | 62.92 | 62.78 | 62.58 | 62.19 | 61.85 |
| | Flow-UniPC | 63.94 | 63.43 | 63.20 | 62.92 | 62.76 | 62.61 | 62.47 | 62.30 | 61.97 | 61.65 |
| | STORK (Ours) | 61.41 | 61.67 | 62.08 | 62.14 | 62.26 | 62.35 | 62.08 | 62.12 | 61.66 | 61.25 |
| Final Score | Flow-DPM-Solver++ | 205 | 211 | 213 | 215 | 216 | 216 | 219 | 218 | 220 | 220 |
| | Flow-UniPC | 206 | 212 | 214 | 216 | 215 | 218 | **219** | **220** | **222** | 221 |
| | STORK (Ours) | **210** | **214** | **215** | **219** | **216** | **218** | 217 | 219 | 220 | **221** |

## H Additional visualizations

We provide additional visualizations. Generated samples for STORK on CIFAR-10 (Krizhevsky et al., 2009) are in Figure 9, and LSUN-Bedroom (Yu et al., 2016) are in Figure 10. Visualizations for MS-COCO-2014 (Lin et al., 2015) are in Figure 11. Visualizations for MJHQ-30K (Li et al., 2024) are in Figure 12 and Figure 13. Visualizations for Hunyuan (Kong et al., 2024) model video generation are in Figure 14 and Figure 15.

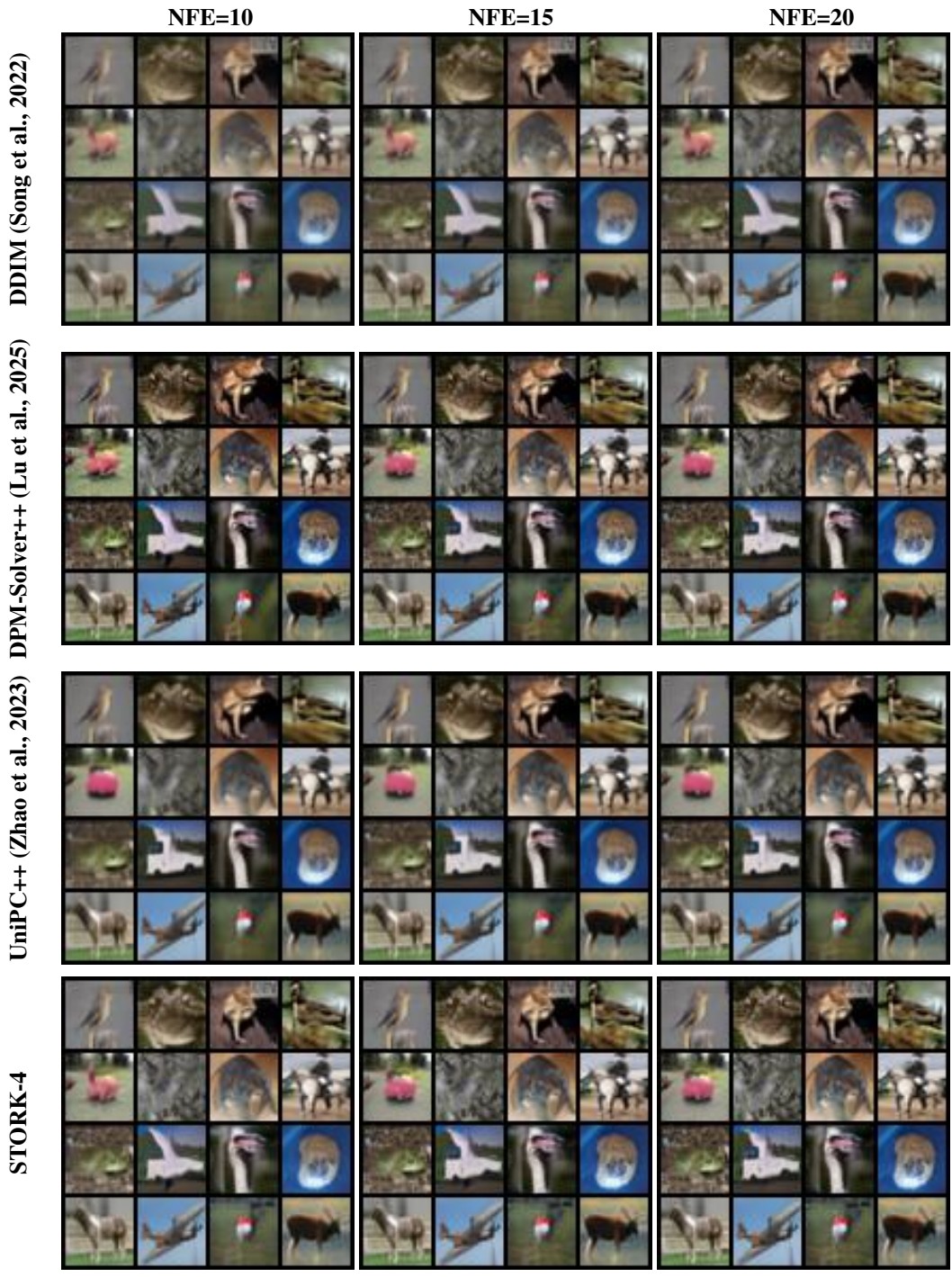

Figure 9: Unconditional generation on CIFAR-10 (Krizhevsky et al., 2009). Generated using DDIM model.

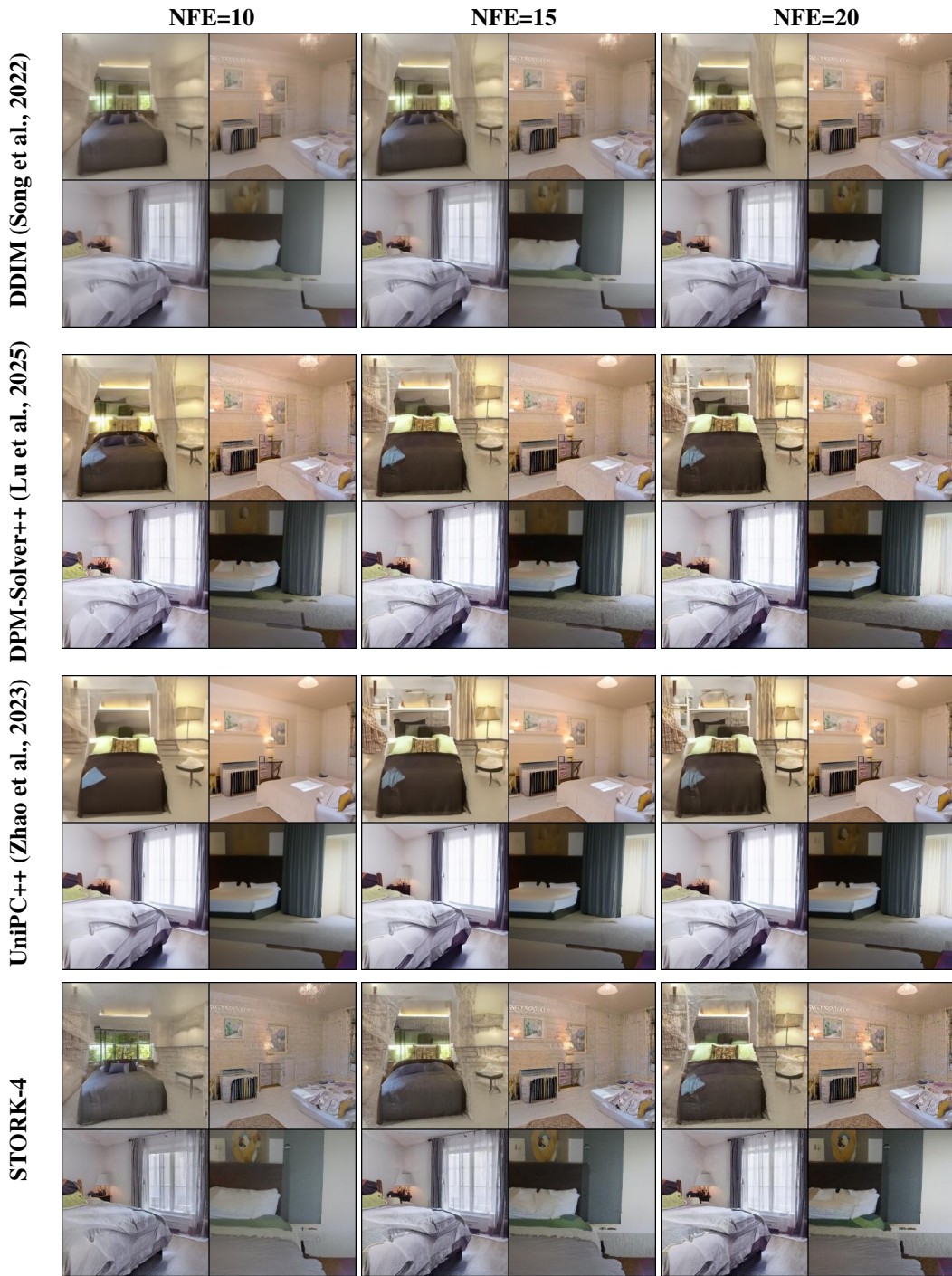

Figure 10: Unconditional generation on LSUN-Bedroom (Yu et al., 2016). Generated using DDIM model.

Flow-Euler          Flow-DPM-Solver++          Flow-UniPC          **STORK (Ours)**

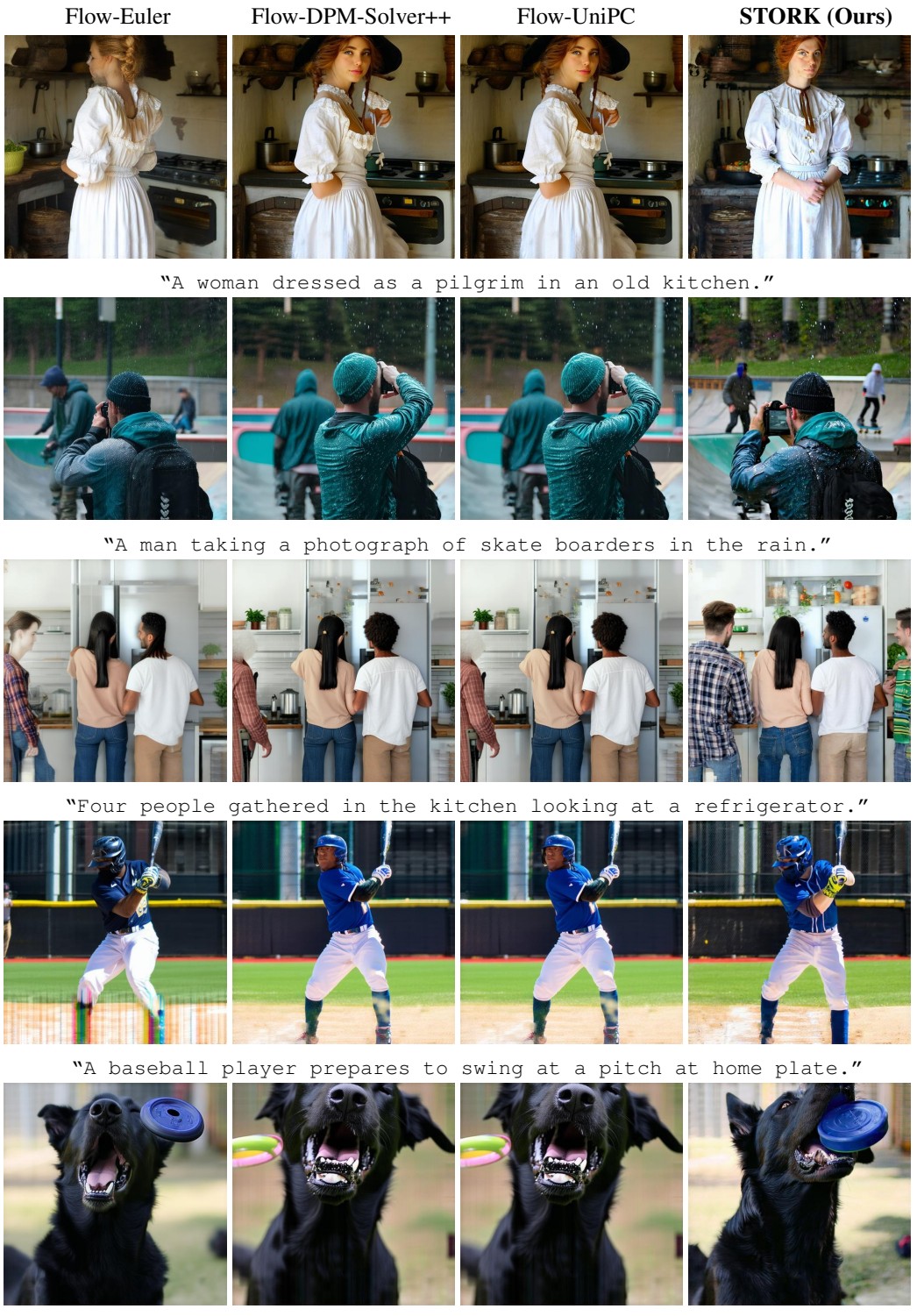

"A woman dressed as a pilgrim in an old kitchen."

"A man taking a photograph of skate boarders in the rain."

"Four people gathered in the kitchen looking at a refrigerator."

"A baseball player prepares to swing at a pitch at home plate."

"A black dog has just captured a Frisbee using its mouth."

Figure 11: Comparison between the Flow-Euler, Flow-DPM-Solver++ (Lu et al., 2025; Xie et al., 2025), Flow-UniPC (Zhao et al., 2023), and STORK. Generated using Stable-Diffusion-3.5-Large (Esser et al., 2024) with 15 NFEs.

| Flow-Euler | Flow-DPM-Solver++ | Flow-UniPC | **STORK (Ours)** |

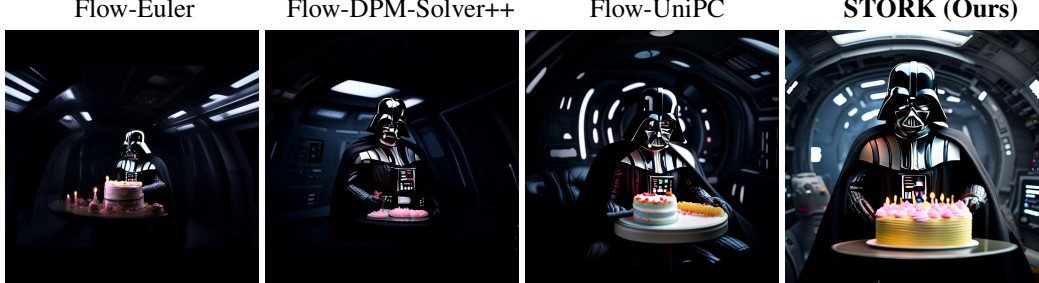

"Darth Vader eating birthday cake in star destroyer interior, cinematic, 50mm". Our image has clearer and lighter background, while other methods have darker backgrounds.

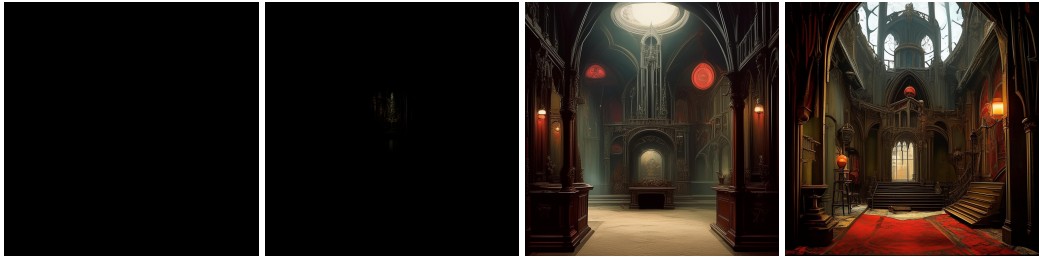

"1920s fantasy interior architecture rebel art, hammer 40K universe, gothic painting". The flow-Euler and flow-DPM-Solver++ fail to generate a decent image, while the flow-UniPC method has worse visualization quality than the STORK method.

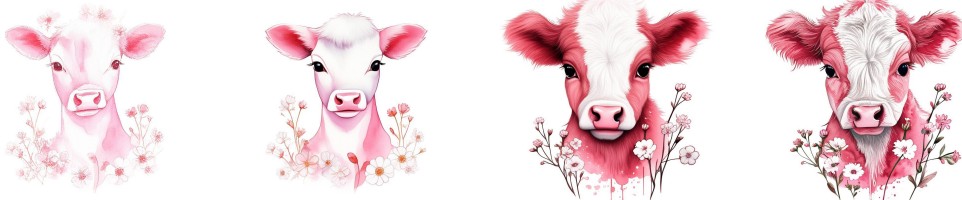

"Watercolor illustration of cute pink and while calf portrait surrounded by small while flowers". Our method has a better image quality and more details than other methods.

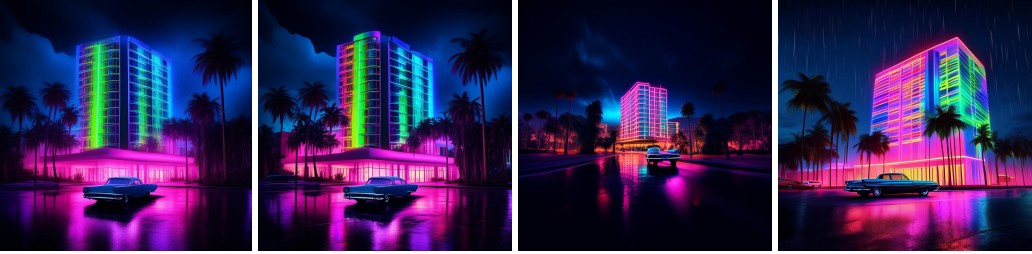

"3D, photorealistic, photo shot, psychedelic, neon colors, neon lights, palm trees, tall, rectangular hotel building, 1960s cars out front, dramatic, dark sky, rain, cinematic, dramatic lighting, hyperrealistic, 8k". Our method clearly shows the rain in the background.

Figure 12: More comparisons between the Flow-Euler, Flow-DPM-Solver++ (Lu et al., 2025; Xie et al., 2025), Flow-UniPC (Zhao et al., 2023), and STORK. All images are generated using the SANA 1.6B model (Xie et al., 2025) at $1024 \times 1024$ resolution with only 8 NFEs, using prompts from MJHQ-30K. Prompts are displayed beneath each image pair, accompanied by our commentary explaining why STORK's generations are superior.

| Flow-Euler | Flow-DPM-Solver++ | Flow-UniPC | **STORK (Ours)** |
|---|---|---|---|

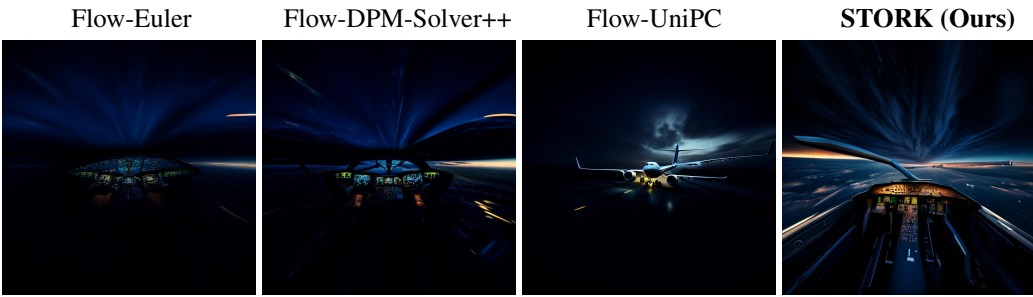

"the cockpit of the Boeing 787 Dreamliner, on a night flight, the night
sky, clouded, dynamic style, depth of Field, F2.8, high Contrast, 8K,
Cinematic Lighting, ethereal light, intricate details, extremely detailed,
incredible details, full colored, complex details, by Weta Digital,
Photography, Photoshoot, Shot on 70mm, real photo, cinematic, high
detailed, HDR, hyper realistic, ".

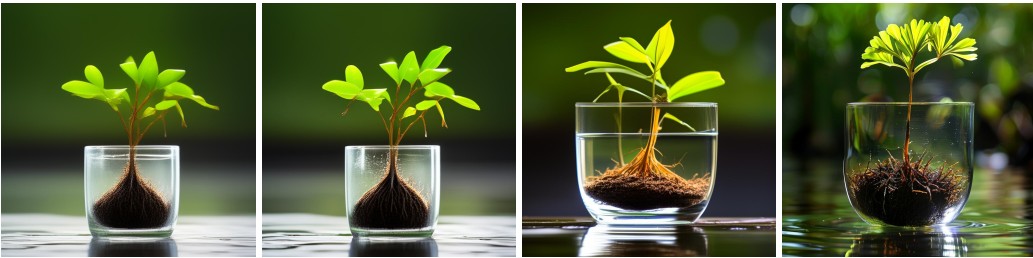

"Mangrove emerging from the seed in glass cup.".

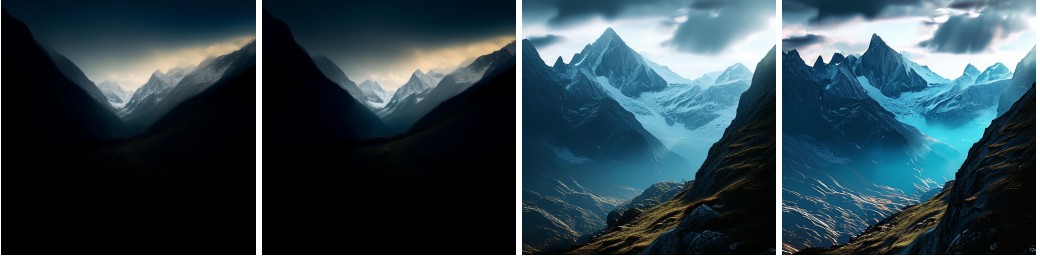

"French Alps, photorealistic, dramatic lighting, mystique scene, ethereal,
majestic, aesthetics, cinematic lighting, deep focus, super adobe,
detailed texture, neuro cognitive art, photoshop, octane render,
Pinterest art, awardwinning landscape photography, world renowned, high
resolution, color grading, high art, no blurs, ultra wideangle lenses,
photo realism, 300 dpi, Ultra Quality, 32k ".

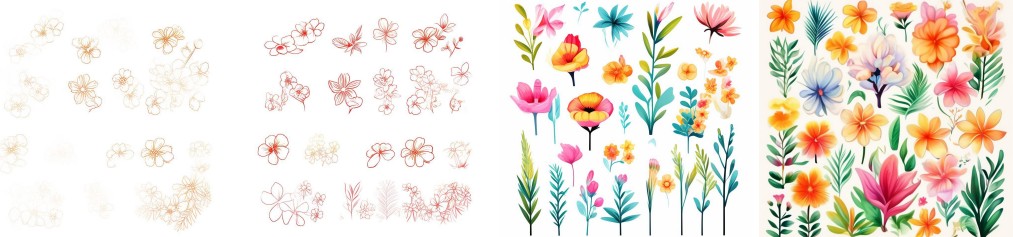

"Paradise summer flowers, clipart, sticker, in the style of aquarellist,
made of flowers, fernando amorsolo, graphic design elements, vector
illustration, in the style of dreamy watercolor florals, floral
explosions, thomas w schaller, sticker art, detailed foliage, cute and
tropical set of a floral pattern, in the style of luminous watercolors
and ink, art deco sensibilities, barbiecore, timeless elegance,
floralpunk, colorful arrangements ".

Figure 13: More comparisons between the Flow-Euler, Flow-DPM-Solver++ (Lu et al., 2025; Xie
et al., 2025)), Flow-UniPC (Zhao et al., 2023)), and STORK on MJHQ-30K, using 8 NFEs.

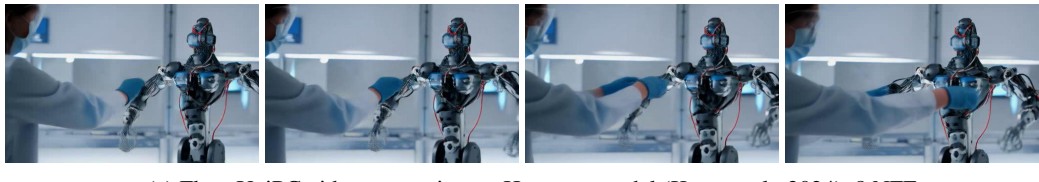

(a) Flow-UniPC video generation on Hunyuan model (Kong et al., 2024), 8 NFE

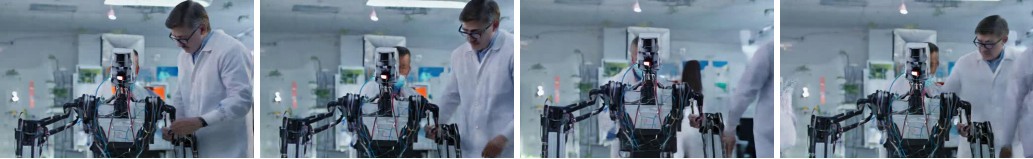

(b) STORK video generation on Hunyuan model (Kong et al., 2024), 8 NFE

Figure 14: Video generation comparison with prompt "`doctors are constructing a robot`".
Our video has more than one doctor, and has more realistic scenario.

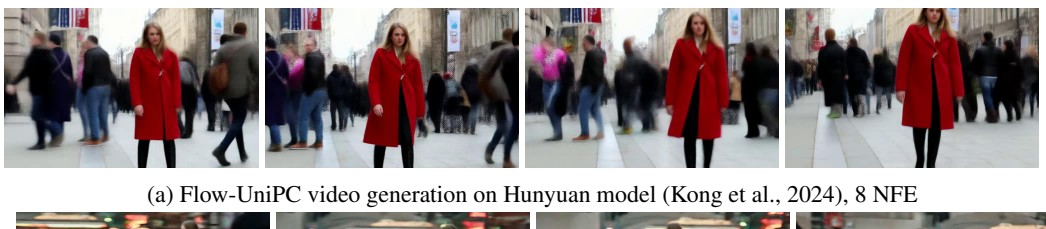

(a) Flow-UniPC video generation on Hunyuan model (Kong et al., 2024), 8 NFE

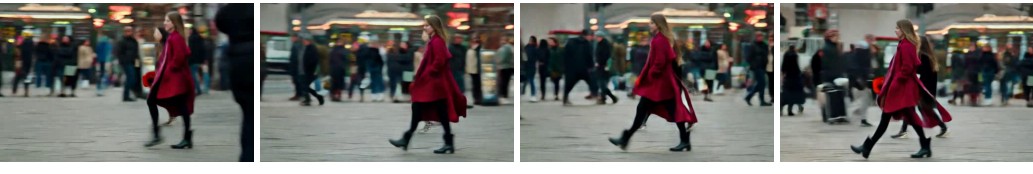

(b) STORK video generation on Hunyuan model (Kong et al., 2024), 8 NFE

Figure 15: Video generation comparison with prompt "`A young woman in her mid-twenties with long blond hair walked briskly down the busy city street, wearing a red coat and black boots`".
Our video has higher quality with a better background.

