# OpenReview forum: "STORK: Faster Diffusion and Flow Matching Sampling by Resolving both Stiffness and Structure-Dependence"
_ICLR.cc/2026/Conference — ICLR 2026 Poster_

### Official Review · Reviewer_VACL · 2025-10-25

**Soundness:** 3
**Presentation:** 3
**Contribution:** 3
**Rating:** 4
**Confidence:** 3

**Summary:**

This paper addresses the challenges of non-straight velocity fields and the dependence on the semi-linear structure in diffusion model ODEs, which make fast but quality-preserving sampling difficult. STROKE tackles these issues by designing a solver that explicitly handles stiffness and structural dependency problems inherent in diffusion and flow-matching models.

**Strengths:**

- The authors redesign the Stable Runge–Kutta (SRK) method to align with the structural characteristics of diffusion and flow dynamics.
- They propose a general framework applicable to both noise-predicting (diffusion-based) and flow-based generative models, extending beyond the semi-linear assumption used in most prior solvers.
- The paper combines rigorous theoretical analysis (including convergence and stability proofs) with comprehensive quantitative and qualitative experiments, demonstrating consistent improvements across different datasets and models.

**Weaknesses:**

- Despite claiming “Stable Runge–Kutta,” the paper does not explore low-NFE (≤5) regimes, where stability advantages would be most pronounced. It remains unclear whether STROKE can outperform existing methods when the step count is drastically reduced.
- The paper lacks comparisons on larger benchmarks such as ImageNet-256 and ImageNet-512, which are standard for evaluating scalability and visual fidelity at higher resolutions.
- Although NFEs are matched across methods, a fair comparison in terms of actual computational cost is missing. The paper introduces the concept of *virtual NFEs*—intermediate approximations computed via Taylor expansion and finite-difference—but does not show how much wall-clock time, memory usage, or throughput efficiency is affected. Internal computation overhead (e.g., Taylor expansion, finite-difference updates, velocity caching) incurs additional cost even without extra NFEs. The paper abstracts this under “virtual NFEs,” but a detailed analysis of real runtime and resource usage would strengthen the claim of efficiency.

**Questions:**

- In Appendix B, the authors illustrate stiffness through a toy example, but can the same stiffness analysis be quantitatively applied to real diffusion or flow-matching models? A visualization or empirical stiffness profile for these models would help support the theoretical motivation.
- In Tables 6 and 7, the baseline outperforms STROKE at smaller NFEs, raising the question: how does STROKE behave in 512×512 image generation when NFE ≤ 5? It would be interesting to see whether SRK’s stability advantages appear under such extreme sampling constraints.

---

> ### Author Response · Authors · 2025-11-19
>
> We are happy to hear that the reviewer found our experimental results comprehensive and consistent. We address the reviewer's individual concerns in the following. We again ask the reviewer to evaluate our paper on the basis of the concrete experimental results.
>
> $$ $$
> **Q1: Stiffness and stability concerns should lead to improvements in extremely low $\le 5$ NFE regime.**
>
> We point out that stability advantages (in the numerical-analysis sense) are **not** most pronounced in the extremely low NFE regime, but rather in medium-to-low NFE regimes **[1]**. In classical numerical analysis, there are two primary concerns that guarantee the convergence of a numerical scheme: consistency and stability; see Definition 1.4.1, 1.4.2, and 1.5.1 in **[2]**. Consistency corresponds to the accuracy of the method, while stability corresponds to the stiffness handling ability of the method. The two concerns are disjoint. Even if a method is unconditionally stable, i.e., it can perfectly handle stiffness, excessively large choice of the timestep can lead the method to fail to converge because of the loss of consistency. In the diffusion-model setting, extremely small NFE corresponds exactly to this loss-of-consistency regime, which is not the regime our method is intended to target.
>
>
> Moreover, we point out that many other training-free sampling methods do not explore the extremely small NFE regime **[3,4]**. Also, in practical use cases, NFEs are not minimized to the extreme for generation. For example, Stable Diffusion 3.5 uses 28 NFEs, FLUX.1-Dev uses 50 NFEs, and SANA uses 20 NFEs for their default settings.
>
> Finally, notice that in our video generation experiments, which is a more difficult task than image generation, we have conducted experiments with NFE starting at 4, and achieve superior performance compared to other methods.
>
>
>
> **Q2: Lack of comparison on larger datasets**
>
> Our image generation experiments, except for CIFAR-10 and LSUN-Bedroom, are all generated on datasets with size $512\times512$ or even with size $1024\times1024$. The strong performance of our method on these datasets demonstrate its ability to achieve high-fidelity generation at large image resolutions.
>
>
> **Q3: Time comparison**
>
> Please refer to the common response.
>
>
> **Q4: Stiffness example on diffusion models**
>
> Please refer to the common response.
>
>
> $$ $$
>
> Given that we positively address main concerns of the reviewer, we kindly ask the reviewer to consider raising the score.
>
> $$ $$
>
>
> [1] R. L. Burden and J. D. Faires. Numerical Analysis. 9th. Boston: Brooks/Cole, 2011, pp. 259–253.
>
> [2] J. C. Strikwerda. Finite Difference Schemes and Partial Differential Equations, Second Edition.
> Society for Industrial and Applied Mathematics, 2004. doi: 10.1137/1.9780898717938. eprint: https://epubs.siam.org/doi/pdf/10.1137/1.9780898717938. url: https://epubs.siam.org/doi/abs/10.1137/1.9780898717938.
>
> [3] C. Lu et al. “DPM-Solver++: Fast Solver for Guided Sampling of Diffusion Probabilistic Models”.
> In: Machine Intelligence Research 22 (Aug. 2025), pp. 730–751. doi: 10.1007/s11633-025-1562-4.
> url: https://doi.org/10.1007/s11633-025-1562-4.
>
> [4] W. Zhao et al. “UniPC: A Unified Predictor-Corrector Framework for Fast Sampling of Diffusion
> Models”. In: Advances in Neural Information Processing Systems (2023).

---

> > ### Comment · Reviewer_VACL · 2025-11-28
> >
> > Thank you for your response.
> >
> > Most of my concerns have been addressed. I will consider adjusting the score.

---

### Official Review · Reviewer_jYva · 2025-10-27

**Soundness:** 2
**Presentation:** 2
**Contribution:** 2
**Rating:** 2
**Confidence:** 4

**Summary:**

The paper introduces STORK, a method for faster sampling from diffusion models by using the Taylor expansion version of a solver (SRK) that is more accurate for stiff ODEs.

**Strengths:**

1. The paper introduces a novel method for faster sampling from diffusion models.
2. The performance of STORK-4 shows improvements when NFE is small.

**Weaknesses:**

1. The paper motivates the proposed method by claiming diffusion models exhibit stiff dynamics, but this is not properly justified empirically or theoretically. There is insufficient evidence presented that diffusion models are actually stiff, making the motivation unclear. Table 1 shows that the STORK-4 is significantly better than SRK4. More intriguingly, even with 50 NFE of SRK4, its performance is around 6.167 for FID score which is lower than its STORK-4 at 10 NFE. This raises the question of whether SRK4 is a good solver for diffusion models and whether the stiffness assumption is valid.
2. Theorem 1 establishes the theoretical guarantee of STORK-4 based on its approximation quality to SRK4. However, it is not clear whether the theory actually supports the empirical finding that 10 NFE of STORK-4 can be better than 50 NFE of SRK4. Additionally, since the goal is to achieve fewer NFE in sampling, does the asymptotic rate (provided in Theorem 1) even matter in practice? The constant in front of the rate may matter more for the low NFE regime that this paper targets. Theorem 1 does not help understand the performance of few-step sampling, leaving the empirical gains unexplained by the theoretical analysis.
3. STORK-4 has many hyperparameters, which makes it unclear whether the observed performance gains come from genuine improvements or from more extensive hyperparameter search. Maybe fix a parameter like s=9 suggested in the paper and report the results for all experiments could help understand the performance gains.

**Questions:**

- Can you clarify why other methods, like the DPM solver, cannot apply to flow matching? What do you mean by the error introduced in each step when you talk about the data prediction step?

---

> ### Author Response · Authors · 2025-11-19
>
> We thank the reviewer for the feedback.
>
> We clarify that our goal is not to rigorously establish stiffness properties in the setting of diffusion models, but rather to present a sampling method with strong and consistent empirical performance. Although, in our view, the superior end-results validate our intuition that guided our theoretical derivation, we believe the experiments should ultimately speak for themselves. Therefore, we again ask the reviewer to evaluate our work primarily on the basis of these concrete experimental results.
>
> $$ $$
>
>
> **Q1: Stiffness assumption**
>
> Please refer to the common response.
>
> **Q2: Purpose of Theorem 1**
>
> The reviewer makes a legitimate point that our order-accuracy theory may not be active in the regime in which the models are ultimately used. On this point, we have two perspectives: (1) The theoretical analysis still provides useful insight to those with a background in numerical analysis into why the method may perform well. (2) For this reason, this type of order analysis is conventionally presented in fast diffusion model sampling methods **[1,2,3]** and in numerical analysis **[4]**.
>
> Our intent in presenting Theorem 1 is to provide soundness and theoretical insight for readers with a numerical analysis background, but we certainly do not claim it to be a main contribution. On that note, we kindly ask the reviewer to evaluate our work primarily on the breadth and strength of our experimental results.
>
>
>
>
> **Q3: Hyperparameter tuning**
>
> We clarify that the single hyperparameter of STORK is just $s$, which controls the number of virtual steps. Appendix G.1 provides extensive experiments on the impact of tuning the single hyperparameter $s$, and we uniformly use $s=9$ for all flow-based models. Other hyperparameters, such as the order of Taylor expansion and the order of the numerical method, are mentioned in the derivation, but are fixed for the noise-based and flow-matching based setting and not tuned in the experiments. Given that we only have one tuning parameter, we confidently claim that the performance gains come from genuine improvements, rather than extensive hyperparameter tuning.
>
> Notice that other papers **[2,3]** also have (arguably much more difficult) hyperparameters to tune, such as the order schedule in UniPC. In particular, Table~4 of the UniPC paper shows that the order parameter in the UniPC method must be tuned carefully and separately for every step.
>
>
>
> **Q4: DPM-Solver for flow matching**
>
> The DPM-Solver relies on the semi-linear structure of the right-hand side of the noise-based diffusion models. It can indeed be adopted into the flow-matching setting [2,5], but doing so comes with a performance cost. As noted in Algorithm 1 of [5], the model-output transformation relies on Tweedie’s formula and introduces stepwise approximation errors. Our formulation avoids this approximation altogether, and our experimental comparison against the flow-adapted version of DPM-Solver shows a clear performance advantage in our favor.
>
> $$ $$
>
>
> Given that we positively address the main concerns of the reviewer, we kindly ask the reviewer to consider raising the score.
>
> $$ $$
>
>
> [1] C. Lu et al. “DPM-Solver: A Fast ODE Solver for Diffusion Probabilistic Model Sampling in Around
> 10 Steps”. In: Advances in Neural Information Processing Systems (2022).
>
> [2] C. Lu et al. “DPM-Solver++: Fast Solver for Guided Sampling of Diffusion Probabilistic Models”.
> In: Machine Intelligence Research 22 (Aug. 2025), pp. 730–751. doi: 10.1007/s11633-025-1562-4.
> url: https://doi.org/10.1007/s11633-025-1562-4.
>
> [3] W. Zhao et al. “UniPC: A Unified Predictor-Corrector Framework for Fast Sampling of Diffusion
> Models”. In: Advances in Neural Information Processing Systems (2023).
>
> [4] R. L. Burden and J. D. Faires. Numerical Analysis. 9th. Boston: Brooks/Cole, 2011, pp. 259–253.
>
> [5] E. Xie et al. “SANA: Efficient High-Resolution Image Synthesis with Linear Diffusion Transform-
> ers”. In: International Conference on Learning Representations (2025).

---

### Official Review · Reviewer_1FHX · 2025-11-01

**Soundness:** 2
**Presentation:** 3
**Contribution:** 3
**Rating:** 4
**Confidence:** 4

**Summary:**

This paper introduces STORK, a new type of training-free numerical ODE solver for diffusion and flow models. STORK is based on SRK methods for solving stiff ODEs. To reduce NFEs, STORK further introduces virtual NFEs, which uses finite difference to approximate the derivatives of the velocity field to extrapolate intermediate velocities without extra NFEs. Experiments across various dataset and pretrained models show that STORK outperforms state-of-the-art ODE solvers such as UniPC and DPM++.

**Strengths:**

- The results presented in this paper are solid. Quantitative evaluations demonstrate a clear advantage of STORK over UniPC and DPM-Solver in few-step settings. Qualitative results also look promising: STORK seems to generate more details than other solvers under few-step settings, especially on video generation.
- Despite the dense math, the presentation quality of this paper is high, making it easily readable. Notably, this paper makes a connection with the notion of stiffness in classical numerical analysis, and then adapts the widely used SRK methods for flow models with a clear motivation.

**Weaknesses:**

- As mentioned in L300, it is stated that "naive application of SRK4 to the CIFAR-10 dataset results in very poor sampling results", so the authors propose to plug Taylor approximation to SRK4. This is an interesting observation, but the reason is not analyzed in depth. If SRK is considered a common method for solving stiff ODEs, why would it not work in this case? Using Taylor expansion and Adams-Bashforth approximation is common in previous flow ODE solvers, so plugging this into SRK seems to make it less unique.
- I wonder how the methods are categorized in Fig. 4. In particular, UniPC, which works very well in practice, is categorized as not stiff, whereas DPM-Solver is considered stiff. Yet it is clear that UniPC can be considered as an extension of DPM-Solver with an additional corrector. Please check this carefully.
- Another claimed advantage of the proposed method is structure-independency. However, in most cases, structure-dependency of ODE solvers is not really an issue in practice. It is very trivial to adopt diffusion solvers using exponential integrators for flow models by rescaling the noise schedule and reparametrizing the prediction format.
- Why is FID getting worse with more NFEs? (fig. 6, Table 6, 10, 11). This behavior is clearly different from DPM-Solver and UniPC, which often converge monotonically. Presenting images generated under higher NFEs could help analyzing this behavior.

**Questions:**

Please see weaknesses.

---

> ### Author Response · Authors · 2025-11-19
>
> We are happy to hear that the reviewer found our results solid and algorithm well motivated. We address the reviewer's individual concerns in the following. We again ask the reviewer to evaluate our paper on the basis of the concrete experimental results.
>
> $$ $$
>
> **Q1: Why vanilla SRK doesn't work**
>
> Please refer to the common response.
>
>
>
> **Q2: Categorization of UniPC**
>
> We clarify that UniPC is not a direct generalization of DPM-Solver. The design objective of DPM-Solver is to utilize the semi-linear structure of the noise/score-based diffusion model probability flow ODE and solve it using a stiff solver, the exponential integrator. In contrast, UniPC is intended to boost the convergence order of the method. In particular, UniPC can be directly inserted as an order booster for any diffusion model sampler, not limited to DPM-Solver. We therefore do not categorize it as a stiff solver.
>
> **Q3: Structure Dependency**
>
> The reviewer is correct that structure-dependent solvers such as DPM-Solver can be adapted to the flow-matching setting **[1,2]**, but doing so comes with a cost. As noted in Algorithm 1 of **[2]**, the model-output transformation relies on Tweedie’s formula and introduces stepwise approximation errors. Our formulation avoids this approximation altogether, and our experimental comparison against the flow-adapted version of DPM-Solver (adopted as the reviewer points out) shows a clear performance advantage in our favor.
>
>
> **Q4: Worse FID with more NFEs**
>
> Thank you for your comment. This is, in fact, a common issue with FID in generative modeling, and many established works report results with non-monotonic FID scores. For example, see for example Fig.3 of Stable Diffusion 3 **[3]**, and Fig.3 and Table 5 of PNDM **[4]**.
>
> We hypothesize that this behavior stems from inherent properties of the FID metric itself, but investigating this shared issue is beyond the scope of our present paper.
>
>
> $$ $$
>
> Given that we positively address the all main concerns of the reviewer, we kindly ask the reviewer to consider raising the score.
>
> $$ $$
>
> [1] C. Lu et al. “DPM-Solver++: Fast Solver for Guided Sampling of Diffusion Probabilistic Models”.
> In: Machine Intelligence Research 22 (Aug. 2025), pp. 730–751. doi: 10.1007/s11633-025-1562-4.
> url: https://doi.org/10.1007/s11633-025-1562-4.
>
> [2] E. Xie et al. “SANA: Efficient High-Resolution Image Synthesis with Linear Diffusion Transform-
> ers”. In: International Conference on Learning Representations (2025).
>
> [3] P. Esser et al. “Scaling Rectified Flow Transformers for High-Resolution Image Synthesis”. In:
> International Conference on Machine Learning (2024).
>
> [4] L. Liu et al. “Pseudo Numerical Methods for Diffusion Models on Manifolds”. In: International
> Conference on Learning Representations (2022).

---

### Official Review · Reviewer_Hn4U · 2025-11-04

**Soundness:** 3
**Presentation:** 2
**Contribution:** 3
**Rating:** 6
**Confidence:** 4

**Summary:**

The paper proposes an advanced ODE sampler, STORK, to handle the stiff dynamics of ODE, while remaining applicable to both score-matching models and flow-matching models. Authors tackle the limitation of prior samplers such as the famous DPM-Solvers, which rely on a semi-linear assumption, thus not natural to be directly applied to flow-matching models. STORK is a variant of stabilized Runge–Kutta (SRK), which is known to handle stiffness well, but it requires expensive intermediate model evaluations. So authors suggest Taylor approximation-based virtual NFEs to approximate internal stages. In experiments, they show strong empirical results across various tasks, including unconditional/conditional image generation and T2V generations, showing consistent improvements under low NFE setups.

**Strengths:**

1. Clear motivation and message: the authors clearly present what problem they aim to solve and how they approach it.
2. Conceptually sound method: Leveraging SRK to handle stiffness, and using Taylor-based virtual NFEs to reduce computational cost, finally making a viable and efficient sampler for flow-matching models
3. Comprehensive empirical evaluation: covering unconditional/conditional image generation and text-to-video tasks, with consistently strong results.
4. Well-written manuscript: the paper is clear and reader-friendly, e.g., additional explanation on Appendix B about stiffness and numerical analysis, parts like this make the paper accessible to broad readers

**Weaknesses:**

1. Runtime analysis: authors are only reporting NFE here, but for a more thorough analysis/comparison, they need to report wall clock time and GPU(VRAM) usage. So that readers could better understand in detail, e.g., how much of that time it takes to virtual NFE calculation?
2. On Table 1,
    - Clarification on NFE report: since they're comparing samplers, with different count evaluations such as inner NFEs for higher-order/intermediate calculation, or virtual NFEs, and so on... Table 1 is a bit hard to read thoroughly and feels unclear.
    The authors should consider explicitly separating NFEs for the super-step/sub-step for clearer comparison.
    - SRK vs STORK: so SRK is super-step (real NFE) + sub-step(real NFE), while STORK is super-step (real NFE) + sub-step(virtual NFE).
    It would be helpful to show the comparison not just in "budget-fair" regime, but in "algorithm-fair" regime, e.g., evaluate both methods with the same super/sub-step structure where SRK operates properly (i.e., with sufficient NFE budget).
    It could strongly back the message “SRK works well but is too expensive, hence we propose STORK to replace those sub-step NFEs with virtual ones.”
3. Stiffness analysis
    - Since the "stiffness" is one of the main motivations, the actual empirical stiffness analysis on pretrained diffusion/flow matching models would strengthen the argument. While calculating Jacobian could be expensive, even a toy-level experiment or approximate analysis would be highly informative and appreciated.

[minor]

1. I understand the limited page for the main part, but it would be beneficial if the ablation table for the main hyperparameter (Table 3 in the supplement), or pseudocode/algorithm (algo1, algo2) could be moved or at least mentioned in the main paper.

**Questions:**

This is a minor question, but Fig5a looks somewhat unusual, where it shows DPM-Solver++ outperforming UniPC, which contradicts to already reported results from other papers - why is this?

---

> ### Author Response · Authors · 2025-11-19
>
> We are happy to hear that the reviewer found our experimental results comprehensive and the algorithm conceptually sound. We address the reviewer's individual concerns in the following. We again ask the reviewer to evaluate our paper on the basis of the concrete experimental results.
>
>
>
> **Responses to the Reviewer's Questions**
>
> **Q1: Run time analysis**
>
> Please refer to the common response.
>
>
> **Q2\&Q3: Table 1 related questions and Stiffness analysis**
>
> Please refer to the common response.  For Table 1 in our paper, we only record the NFE counts for comparison since we aim to do so as an ablation study across different fourth-order sampling methods; similar experiments can be found in Table 1 of **[1]**. For a clearer comparison for SRK4 and STORK-4 with the same super-step and sub-step structure, we conduct the experiments using $s$=14 on CIFAR for SRK4 and STORK-4 as the same setting in the paper, and compare between different methods. As shown in Table A, the SRK4 achieves lower or comparable results as the STORK-4 when the NFE bugdet is sufficient, so that we use virtual NFE to reduce it to STORK-4 for a much lighter NFE cost. Since SRK4 is a stiff solver by itself **[2]**, this also empirically shows the stiffness property of the diffusion models.
>
>
> **Q4: UniPC versus DPM-Solver++**
>
> We compared all the baseline methods using the implementation on the HuggingFace Diffuser. Although on CIFAR-10 the UniPC method performs worse compared to the DPM-Solver++ in this setting, in most of the other datasets and models the UniPC is better than DPM-Solver++ under the same parameter setting as expected.
>
> **Minor concern**
>
> We agree with the reviewer’s suggestion. Upon acceptance, we will move the mentioned content into the main text using the additional page given for the final camera-ready version.
>
> $$ $$
>
> Given that we positively address the main concerns of the reviewer, we kindly ask the reviewer to consider raising the score.
>
> [1] C. Lu et al. “DPM-Solver: A Fast ODE Solver for Diffusion Probabilistic Model Sampling in Around
> 10 Steps”. In: Advances in Neural Information Processing Systems (2022).
>
> [2] A. Abdulle. “Fourth Order Chebyshev Methods with Recurrence Relation”. In: SIAM Journal
> on Scientific Computing 23.6 (2002), pp. 2041–2054. doi: 10.1137/S1064827500379549. eprint:
> https://doi.org/10.1137/S1064827500379549. url: https://doi.org/10.1137/S1064827500379549.

---

### Author Response · Authors · 2025-11-19

The reviewers generally found our methodology to be novel and our experiments comprehensive, but raised concerns about:

(1) Whether virtual NFEs introduce additional computational cost, potentially making comparisons unfair.

(2) The motivation and derivation involving stiffness and the SRK method.

We address these concerns in the common response below.

That said, we believe that **the concrete experimental results constitute our main contribution, and these numbers should speak for themselves**. We conduct extensive experiments across a wide range of settings, including both text-to-image and text-to-video generation. The gains brought by our sampler are significant and consistent. In addition, our implementation already conveniently interfaces with the widely used Huggingface Diffusers library, allowing the broader research community to easily adopt our sampler and benefit from it, thereby providing large aggregate value.

Therefore, **we ask the reviewers to evaluate our work on the basis of the experimental results.**




## Runetime analysis: Wall-clock time
We measure the inference time of baselines methods and STORK on diffusion and flow-matching models using 10 NFEs, averaged over 10 trials. For STORK, we use `s=9`, which is the choice of hyperparameter reported used in our paper, meaning 9 "virtual steps" are performed for each NFE. All experiments are conducted on a single H100-NVML GPU and generating images using `batch_size=1` with `FLUX.1-Dev` loaded in `torch.bfloat16` generating an at `(512, 512, 3)` resolution.




**Table A.** Inference walltime profiling across different flow-matching sampling methods all with 10 NFEs
| Solver   | Avg (s) | Min (s) | Max (s)
| -------- |:-------:|:-------:| :-------:|
| Flow-DPM-Solver++ **[1]** | 1.100  | 1.042  | 1.583
| Flow-UniPC **[2]**        | 1.223  | 1.167  | 1.699
| STORK              | 1.224  | 1.167  | 1.687


As shown, the computation overhead incurred by the virtual steps is negligible.



$$ $$

## Runtime analysis: Memory consumption
At each step, at most one previous velocities are needed when using the first-order Taylor expansion approximation, and this extra cost is essentially negligible. For latent space diffusion,a typical latent representation has size $64\times64\times256$; using the `torch.bfloat16` datatype, this amounts to $1\times64\times64\times256\times2=2,097,152\text{ bytes}=\textbf{2 MiB}$. Additionally, note that storing previous computation values is common in established sampling methods such as PNDM **[3]** and DPM-Solver++ **[1]**.


Following the hardware (H100-NVML) and model setups (`FLUX.1-Dev` using `torch.bfloat16`) from the wall-clock time measurements, we report the following results.


**Table B.** Memory profiling across different flow-matching sampling methods
| Solver   | Max GPU-Mem Allocated | Max GPU-Mem Reserved
| -------- |:-------:|:-------:|
| Flow-DPM-Solver++  | 32.069 GB | 32.580 GB
| Flow-UniPC         | 32.069 GB | 32.580 GB
| STORK              | 32.071 GB | 32.584 GB

As shown, the GPU memory usage is almost identical, validating our analysis.



$$ $$


## Stiffness and SRK

Vanilla SRK4 is evidently a poor sampling method, as it requires an excessively large number of NFEs, as shown in `Table 1` of the paper. Nonetheless, some reviewers expressed curiosity about how SRK4 would perform if it were given the same number of super-steps as STORK-4 (even though this would result in SRK4 using far more NFEs than its STORK-4 counterpart).

In the following `Table C`, we report experimental results on CIFAR-10 using $s=14$ for both SRK4 and STORK-4 with the the setting used in the paper, and compare their performance also against the other baselines. The strong performance of SRK4 (in this unrealistic setup) empirically validates the stiffness concerns that motivated the design of STORK.


**Table C.** FID $\downarrow$ comparison of different methods with same super-step structure.
| Superstep count   | 8 | 9 | 10 | 12 |
| -------- |:-------:|:-------:| :-------:| :-------:|
| DDIM | 23.260 | 20.390 | 18.500 | 15.477 |
| UniPC | 19.732 | 17.879 | 16.666 | 14.593|
| DPM-Solver++ | 8.669 | 7.250 | 6.471 | 5.525 |
| SRK4    | **5.429**  | **5.447**  | **5.250** | 5.080 |
| STORK-4  | 6.753  | 5.743  | 5.497 | **4.964** |



## Reference
[1] C. Lu et al. “DPM-Solver++: Fast Solver for Guided Sampling of Diffusion Probabilistic Models”.
In: Machine Intelligence Research 22 (Aug. 2025), pp. 730–751. doi: 10.1007/s11633-025-1562-4.
url: https://doi.org/10.1007/s11633-025-1562-4.


[2] W. Zhao et al. “UniPC: A Unified Predictor-Corrector Framework for Fast Sampling of Diffusion
Models”. In: Advances in Neural Information Processing Systems (2023).


[3] L. Liu et al. “Pseudo Numerical Methods for Diffusion Models on Manifolds”. In: International
Conference on Learning Representations (2022).

---

### Author Response · Authors · 2025-12-01
**Quick Message to the New AC**

Although the reviewers generally agree on the breadth and strength of our experimental results, we find that the reviewers are somewhat distracted by our derivation of the method and the intuition surrounding stiffness. We believe the concrete fast sampling experimental results constitute our main contribution, and kindly ask the AC to evaluate our paper with attention to the following points:

1. Our experiments consistently demonstrate strong and solid gains compared with existing methods.

2. The experimental evaluation is comprehensive, spanning a range of datasets and tasks.

3. Our response satisfied Reviewer VACL, as reflected in the review record. (The discussion period was suspended before any of the other reviewers could respond.)

---

### Meta-Review · Area_Chair_sU73 · 2026-01-06

**Summary:**

The reviewers were concerned about the lack of evidence for the stiffness of diffusion models, missing runtime analysis, the discrepancy between theory and practice, and not considering the extremely low NFE regime.

While there are still outstanding concerns after the rebuttal, this paper adapts a well-established numerical solver to accelerate diffusion models and flow matching with fewer NFEs. The experiments for multiple SOTA models with high image resolution are compelling, and the proposed training-free method is indeed practically useful for the community. Hence, the AC recommends accepting this paper.

**Reviewer Concerns:**

Concerns that were addressed by the rebuttal:
- Runtime and memory usage analysis (reviewer Hn4U, VACL). The authors have reported latency and GPU memory usage. The proposed solver doesn't incur much computational overhead.
- Comparison to SRK4 with the same number of super-steps. It is not clear why SRK doesn't work (reviewer Hn4U, 1FHX). The rebuttal includes experiments to compare SRK4 and SRORK-4 with the same number of super-steps.
- It is a trivial adaptation of diffusion solvers using exponential integrators for flow models. (Reviwer 1FHX).
- Hyperparameter tuning for STORK. (Reviwer jYva)


Outstanding concerns:
- The stiffness assumption of diffusion models is not justified empirically or theoretically. (Reviwer jYva)
- Discrepency between the theory on the convergence of STORK-k and the practical observation. (Reviewer jYva)
- Not considering extremely low NFE regime (<5) (Reviewer VACL)

**Reviewer Scores:**

Reviewer Hn4U is likely to maintain a rating of 6.

Reviewer 1FHX is likely to raise the score from 4 to 6.

Reviewer jYva is likely to keep a score of 2.

Reviewer VACL has indicated that most concerns were addressed. Reviewer vaCL is likely to increase rating from 4 to 6.

---

### Decision · Program_Chairs · 2026-01-26

Accept (Poster)